# Visual-Based SLAM Configurations for Cooperative Multi-UAV Systems with a Lead Agent: An Observability-Based Approach

**DOI:** 10.3390/s18124243

**Published:** 2018-12-03

**Authors:** Juan-Carlos Trujillo, Rodrigo Munguia, Edmundo Guerra, Antoni Grau

**Affiliations:** 1Department of Computer Science, CUCEI, University of Guadalajara, Guadalajara 44430, Mexico; juancarlos_max@hotmail.com; 2Department of Automatic Control, Technical University of Catalonia UPC, 08034 Barcelona, Spain; edmundo.guerra@upc.edu (E.G.); antoni.grau@upc.edu (A.G.)

**Keywords:** cooperative SLAM, aerial robots, state estimation, observability

## Abstract

In this work, the problem of the cooperative visual-based SLAM for the class of multi-UA systems that integrates a lead agent has been addressed. In these kinds of systems, a team of aerial robots flying in formation must follow a dynamic lead agent, which can be another aerial robot, vehicle or even a human. A fundamental problem that must be addressed for these kinds of systems has to do with the estimation of the states of the aerial robots as well as the state of the lead agent. In this work, the use of a cooperative visual-based SLAM approach is studied in order to solve the above problem. In this case, three different system configurations are proposed and investigated by means of an intensive nonlinear observability analysis. In addition, a high-level control scheme is proposed that allows to control the formation of the UAVs with respect to the lead agent. In this work, several theoretical results are obtained, together with an extensive set of computer simulations which are presented in order to numerically validate the proposal and to show that it can perform well under different circumstances (e.g., GPS-challenging environments). That is, the proposed method is able to operate robustly under many conditions providing a good position estimation of the aerial vehicles and the lead agent as well.

## 1. Introduction

In recent years, advances in technology and miniaturization of flight control systems have contributed to the growth of interest for the Unmanned Aerial Vehicles (UAVs). The UAVs are very versatile in their movements making them suitable for a large number of applications [1,2]. For many applications, it is necessary that two or more UAVs perform a flight formation with respect to a lead agent, which can be another UAV, a human, or another kind of mobile robot or vehicle (multi-UAV systems with a lead agent). In general, in these kinds of systems, the lead agent will be the only member of the formation that can move freely through its environment. To control the flight formation, it is necessary to have knowledge about the states (localization) of the aerial robots as well as the state of the lead agent.

In different scenarios, autonomous navigation capability is a primary mission requirement, which in outdoor environments it is typically fulfilled by integrating a Global Positioning System (GPS). However, several mission profiles require a multi-UAV system to operate in GPS-challenging or GPS-denied environments. In these kinds of environments (e.g., natural or urban canyons, or mixed indoor-outdoor scenarios) multipath or shadowing of the GPS satellite signal creates serious difficulties for GPS receivers to yield a reliable position. Therefore, a sort of additional sensory information (e.g., visual information) should be integrated into the system in order to improve accuracy and robustness. In a practical scenario, it is clear that during a flight mission there can exist periods of time where the whole set of (or some) UAVs moves indistinctly through a GPS-challenging or a GPS-denied environment.

In the above context, more works have appeared focusing on the use of cameras in order to develop navigation systems based on visual information that can operate when GPS is partially available or denied. Moreover, even when the position sensor is available, the visual information can be fused in order to improve the accuracy and robustness of the system. Also, cameras are well suited for their use in embedded systems. In this work, the use of a cooperative visual-based SLAM (Simultaneous Localization and Mapping) scheme is studied for addressing the problem of estimating at the same time the location of both the aerial robots and the lead agent. The general idea is that the aerial vehicles, flying in formation with respect to the lead agent, carry out the task of self-localization as well as locating the lead agent. The above scheme allows flexibility and freedom for the lead agent to carry out its own mission without the restriction of fulfilling the task of localizing itself (See Figure 1).

### 1.1. Related Work

The visual-based SLAM methods make use of visual features as landmarks and can be used for addressing the problem of the state estimation of robots. In this scenario, the robot operates in a priori unknown environment using only angular measurements obtained from cameras, to simultaneously building a map of its surroundings which it is used at the same time to track its position. Currently, there are two main approaches for implementing vision-based SLAM systems: (i) Filtering-based methods ([3,4,5]) and (ii) the optimization-based methods ([6,7]).

In the literature, there are several works that present SLAM methods applied to UAVs, such as in [8], where a Visual SLAM method is developed in outdoors, in partially structured environments. In [9], the estimation of the UAV state obtained through a Visual SLAM method is used as feedback to the control laws that stabilize the aircraft. In [10], a Visual SLAM method is developed for GPS-challenging environments. In [11], the estimation of the UAV state obtained through a Visual SLAM method is used to carry out emergency landings. In [12], a sensor fusion method for Visual SLAM through the integration of a monocular camera and a 1D-laser range-finder is presented. In [13], an approach based on graph-SLAM and loop closure detection for online mapping of unknown outdoor environments, using a small UAV, is proposed. In [14], a Inertial-Visual SLAM is proposed. In this method, inertial measurements are integrated into the system in order to recover the metric scale of the estimations. On the other hand, the performance of this method can be affected by the dynamic error bias which is common to low-cost MEMS sensors. In [15], an EKF-based method is proposed in order to perform visual odometry with an unmanned aircraft. This method makes use of inertial sensors, a monocular downward facing camera and a range sensor (sonar altimeter). Unlike vision-based SLAM, in visual odometry approaches, there is not a mapping process. Furthermore, in those approaches, the operating altitude of the UAV is limited by the operating range of the sonar. More recently, another approach that addresses the problem of visual-based navigation in GPS-denied environments ([16,17,18]) has been appearing.

Multi-robot systems have also received great attention from the robotics research community. This attention is motivated by the inherent versatility that those systems are performing tasks that could be difficult to realize by a single robot. The use of several robots can have advantages like cost reductions, more robustness, better performance and efficiency ([19,20]). In the case of the SLAM problem, in [21,22] a centralized architecture is used where all vehicles send their sensor data to a unique Kalman filter. In [15,23,24] the idea of combining monocular SLAM with cooperative, multi-UAV information to improve navigation capabilities in GPS-challenging environments is presented. In [25], the idea of combining monocular SLAM with cooperative, human-robot information to improve navigation capabilities is presented. In [26], a visual-based cooperative localization method is proposed. According to the analysis presented in this work, the proposed system is completely observable. However, in this case, only distances and the relative orientations between the robots are estimated. This fact can be a clear drawback for applications where the global measurements of the system are required (e.g., absolute position).

As will be seen later, an important UAV technology that it is needed in order to implement the architectures proposed in this work is the cooperative visual-based detection and tracking of mobile targets. In this case, there are several works that deal with these kinds of problems. For instance, [27] presents a vision-based target detection and localization system, that makes use of different capabilities of aerial and ground unmanned vehicles as a cooperative team. In [28], a visual-based approach that allows a UAV to detect and track a cooperative flying vehicle autonomously using a monocular camera is presented. The algorithms are based on template matching and morphological filtering. In [29], a sophisticated vision-aided flocking system for UAVs is presented, which is able to operate in GPS-denied unknown environments, for missions of exploring and searching. In [30], it is presented a cooperative vision-based estimation and tracking system for objects of interest that are located on or near the ground. In the work in [31], an automatic cooperative tracking of targets system, using two quadrotors UAVs equipped with stereo vision systems, is presented. The system includes vision-based algorithms for searching and detecting of the target on the video stream. The research in [32] addresses the topic of vision-based target detection and tracking using a team of UAVs for maritime border surveillance. In this work, a method on how to integrate the perception into the control loop using two distinct teams of UAVs that are cooperatively tracking the same target is presented.

### 1.2. Objectives and Contributions

In this work, three configurations of cooperative SLAM for multi-UAV systems containing a lead agent are presented and analyzed. In all cases, each aerial robot is equipped with a monocular camera on board. The main differences between the system configurations have to do with the set of sensors used in each case (additional to the monocular cameras), as well as the circumstances they operate in a suitable manner.

The idea of presenting different system configurations, instead of a single ”integrated” proposal, has to do the easiness of the mathematical analysis. However, also, this will allow a higher flexibility and modularity in the system implementation, since the features of each configuration are complementary and not exclusive. In other words, the authors’ proposal should facilitate the implementation of an integrated system according to requirements (or availability) of hardware and the circumstances where the system will operate.

The configurations of the cooperative SLAM system proposed in this paper are based on the standard EKF-based SLAM methodology. In this context, it is extremely important to provide the proposed system with properties such as observability and robustness to errors in initial conditions since the properties mentioned above have a fundamental role in the convergence of the filter, as shown in [33,34]. Therefore, an extensive nonlinear observability test is carried out in order to analyze the system configurations. In this case, novel and important theoretical results and considerations are presented from the observability analysis. Also, in this work, the initialization process of new map features is carried out in a cooperative way. The 3D position of the new map features is estimated by means of a pseudo-stereo system formed by monocular cameras mounted on a pair of UAVs that observe common landmarks. This allows the landmark to be initialized with less uncertainty and lesser error. The pseudo-stereo system allows to initialize landmarks at distances farther than using stereo systems with a rigid baseline [35] or delayed monocular initialization methods. The above feature allows to the proposed cooperative system to have better performance in environments where the landmarks are far from the measurement system, contrary to SLAM approaches based on depth cameras, standard stereo systems or sonars.

### 1.3. Paper Outline

The document is organized in the following manner: Section 2 presents the general system specifications and mathematical models. Section 3 introduces the system configurations proposed in this work. Section 4 presents the nonlinear observability analysis. In Section 5 the proposed method is described. Section 7 shows the numerical simulations results, and finally, in Section 8 the conclusions of this work are presented.

## 2. System Specification

In this section, the different mathematical models used in this work are introduced. Those models are: the model used for representing the dynamics of a camera carried by a quadcopter, the model used for representing the dynamics of the lead agent, the representation of the landmarks as map features, the camera projection, and the GPS, altimeter, and range measurement models.

### 2.1. Dynamics of the System

In applications like aerial vehicles, the attitude and heading (roll, pitch, and yaw) estimation is well handled by available systems (e.g., [36,37]). In particular, in this work is assumed that the orientation of the camera always points toward the ground. In practice, the foregoing assumption can be easily addressed with the use of a servo-controlled camera gimbal. Considering the above aspects, the system state can be simplified by removing the variables related to attitude and heading which are provided by the attitude and heading reference system (AHRS). Therefore, the problem will be focused on the position estimation. Let consider the following continuous-time model describing the dynamics of the proposed system (see Figure 2):(1)$$\dot{x}=\left[\begin{array}{c} {\dot{x}}_{h}\\ {\dot{v}}_{h}\\ {\dot{x}}_{c}^{j}\\ {\dot{v}}_{c}^{j}\\ {\dot{x}}_{a}^{i} \end{array}\right]=\left[\begin{array}{c} \;\;\;\;v_{h}\\ \;\;\;0_{3\times 1}\\ \;\;\;\;v_{c}^{j}\\ \;\;\;0_{3\times 1}\\ \;\;\;0_{3\times 1} \end{array}\right]$$
where the state vector x is defined by:(2)$$x={\left[\begin{array}{ccccc} x_{h} & v_{h} & x_{c}^{j} & v_{c}^{j} & x_{a}^{i} \end{array}\right]}^{T}$$
with $i=1,\ldots ,n_{1}$ and $j=1,\ldots ,n_{2}$, where $n_{1}$ and $n_{2}$ are respectively the number of landmarks included into the map and the number of UAV-camera systems. In this work, the term *landmarks* will be used to refer to natural features of the environment which are detected and tracked from the images acquired by the cameras.

Additionally, let $x_{h}={\left[\begin{array}{ccc} x_{h} & y_{h} & z_{h} \end{array}\right]}^{T}$ represent the position (in meters) of the lead agent, with respect to the reference system *W*. Let $x_{c}^{j}={\left[\begin{array}{ccc} x_{c}^{j} & y_{c}^{j} & z_{c}^{j} \end{array}\right]}^{T}$ represent the position (in meters) of the reference system *C* of the *j*-th camera, with respect to the reference system *W*. Let $v_{h}={\left[\begin{array}{ccc} {\dot{x}}_{h} & {\dot{y}}_{h} & {\dot{z}}_{h} \end{array}\right]}^{T}$ represent the linear velocity (in $\frac{m}{s}$) of the lead agent. Let $v_{c}^{j}={\left[\begin{array}{ccc} {\dot{x}}_{c}^{j} & {\dot{y}}_{c}^{j} & {\dot{z}}_{c}^{j} \end{array}\right]}^{T}$ represent the linear velocity (in $\frac{m}{s}$) of the *j*-th camera. Finally, let $x_{a}^{i}={\left[\begin{array}{ccc} x_{a}^{i} & y_{a}^{i} & z_{a}^{i} \end{array}\right]}^{T}$ be the position of the *i*-th landmark (in meters) with respect to the reference system *W*, defined by its Euclidean parameterization. In (Equation 1), each UAV-camera, as well as the lead agent, is assumed to move freely in the three-dimensional space. Let note that a non-acceleration model is assumed for the UAV-camera systems and the lead agent. Also note that the landmarks are assumed to remain static.

### 2.2. Camera Measurement Model for the Projection of the Landmarks

Let consider the projection of a single landmark over the image plane of a camera. Using the pinhole model [38] (see Figure 3) the following expression can be defined:(3)$$\begin{array}{c} {}^{i}z_{c}^{j}={\;}^{i}h_{c}^{j}=\left[\begin{array}{c} {}^{i}u_{c}^{j}\\ {}^{i}v_{c}^{j} \end{array}\right]=\frac{1}{{}^{i}z_{d}^{j}}\left[\begin{array}{cc} \frac{f_{c}^{j}}{d_{u}^{j}} & 0\\ 0 & \frac{f_{c}^{j}}{d_{v}^{j}} \end{array}\right]\left[\begin{array}{c} {}^{i}x_{d}^{j}\\ {}^{i}y_{d}^{j} \end{array}\right]+\left[\begin{array}{c} c_{u}^{j}+d_{ur}^{j}+d_{ut}^{j}\\ c_{v}^{j}+d_{vr}^{j}+d_{vt}^{j} \end{array}\right] \end{array}$$

Let ${[}^{i}u_{c}^{j}$, ${}^{i}v_{c}^{j}]$ define the coordinates (in pixels) of the projection of the *i*-th landmark over the image of the *j*-th camera. Let $f_{c}^{j}$ be the focal length (in meters) of the *j*-th camera. Let $\left[d_{u}^{j},d_{v}^{j}\right]$ be the conversion parameters (in m/pixel) for the *j*-th camera. Let $\left[c_{u}^{j},c_{v}^{j}\right]$ be the coordinates (in pixels) of the image central point of the *j*-th camera. Let $\left[d_{ur}^{j},d_{vr}^{j}\right]$ be components (in pixels) accounting for the radial distortion of the *j*-th camera. Let $\left[d_{ut}^{j},d_{vt}^{j}\right]$ be components (in pixels) accounting for the tangential distortion of the *j*-th camera. All the intrinsic parameters of the *j*-th camera are assumed to be known by means of some calibration method. Let ${}^{i}p_{d}^{j}={\left[\begin{array}{ccc} {}^{i}x_{d}^{j} & {}^{i}y_{d}^{j} & {}^{i}z_{d}^{j} \end{array}\right]}^{T}$ represent the position (in meters) of the *i*-th landmark with respect to the coordinate reference system *C* of the *j*-th camera.

Additionally,
(4)$${}^{i}p_{d}^{j}={}^{W}R_{c}^{j}\left(x_{a}^{i}-x_{c}^{j}\right)$$
where ${}^{W}R_{c}^{j}\in SO3$ is the rotation matrix, that transforms from the world coordinate reference system *W* to the coordinate reference system *C* of the *j*-th camera. Recall that the rotation matrix ${}^{W}R_{c}^{j}$ is known and constant, by the assumption of the use of the servo-controlled camera gimbal.

### 2.3. Camera Measurement Model for the Projection of the Lead Agent

Let consider the projection of the lead agent over the image plane of a camera. In this case, it is assumed that some visual feature points can be extracted from the lead agent by means of some available computer vision algorithm like [39,40,41,42] or [43].

Using the pinhole model (see Figure 3) the following expression can be defined:(5)$$\begin{array}{c} {}^{h}z_{c}^{j}={\;}^{h}h_{c}^{j}=\left[\begin{array}{c} {}^{h}u_{c}^{j}\\ {}^{h}v_{c}^{j} \end{array}\right]=\frac{1}{{}^{h}z_{d}^{j}}\left[\begin{array}{cc} \frac{f_{c}^{j}}{d_{u}^{j}} & 0\\ 0 & \frac{f_{c}^{j}}{d_{v}^{j}} \end{array}\right]\left[\begin{array}{c} {}^{h}x_{d}^{j}\\ {}^{h}y_{d}^{j} \end{array}\right]+\left[\begin{array}{c} c_{u}^{j}+d_{ur}^{j}+d_{ut}^{j}\\ c_{v}^{j}+d_{vr}^{j}+d_{vt}^{j} \end{array}\right] \end{array}$$

Let ${}^{h}p_{d}^{j}={\left[\begin{array}{ccc} {}^{h}x_{d}^{j} & {}^{h}y_{d}^{j} & {}^{h}z_{d}^{j} \end{array}\right]}^{T}$ represent the position (in meters) of the lead agent with respect to the coordinate reference system *C* of the *j*-th camera.

Additionally,
(6)$${}^{h}p_{d}^{j}={}^{W}R_{c}^{j}\left(x_{h}-x_{c}^{j}\right)$$

### 2.4. GPS Measurement Model

Let consider a GPS carried by the lead agent. From the GPS, measurements of the global position of the lead agent are obtained, therefore it can be defined:(7)$$z_{g}=h_{g}=x_{h}$$

### 2.5. Altimeter Measurement Model

Let consider an altimeter carried by the *j*-th quadcopter. From the altimeter, measurements of the altitude of the *j*-th quadcopter are obtained, therefore it is defined:(8)$$z_{a}^{j}=h_{a}^{j}=z_{c}^{j}$$

### 2.6. Range Measurement Model

Let consider a range sensor, from which measurements of the relative distance of the *j*-th quadcopter respect to the lead agent can be obtained. Therefore, it is defined:(9)$${}^{h}z_{r}^{j}={\;}^{h}h_{r}^{j}=\sqrt{{\left(x_{h}-x_{c}^{j}\right)}^{2}+{\left(y_{h}-y_{c}^{j}\right)}^{2}+{\left(z_{h}-z_{c}^{j}\right)}^{2}}$$

For a practical implementation, several techniques like [44] or [45] can be used in order to obtain these kinds of range measurements.

## 3. System Configurations

In this section, three different configurations of cooperative visual-based SLAM for Multi-UAV systems with a lead agent are introduced:

### 3.1. First Configuration

The first configuration takes into account the possibility of visual contact of at least one of the robots with the lead agent (See Figure 1). The system in its first configuration uses as inputs the following measurements: (i) the measurements obtained of the GPS carried by the lead agent, (ii) the monocular measurements obtained from the projection of the landmarks over each UAV-camera system, and (iii) the monocular measurements obtained from the projection of the lead agent over an UAV-camera system that has visual contact with it.

### 3.2. Second Configuration

The second configuration takes into account a scenario where there is no visual contact from any robot with the lead agent. In this case, the system is assumed to have the following sensor measurements: (i) the measurements obtained of the GPS carried by the lead agent, (ii) the monocular measurements obtained from the projection of the landmarks over each UAV-camera system, (iii) the measurements of the range of a robot with respect to the lead agent, and iv) the measurements obtained of an altimeter carried by one of the robots.

### 3.3. Third Configuration

The third configuration takes into account the scenarios where the GPS measurements are unavailable (GPS-denied environments). In Section 4, it will be analysed that in the absence of GPS the system will be partially observable for any possible configuration. Therefore, the objective will be to reduce the number of unobservable variables and modes of the system. In this case, in order to maximize the observability property of the system, in the absence of GPS, the third configuration is obtained by combining the first two configurations. The system in its third configuration is composed of the following measurements: (i) the monocular measurements obtained from the projection of the landmarks over each UAV-camera system, (ii) the monocular measurements obtained from the projection of the lead agent over an UAV-camera system that has visual contact with it, (iii) the measurements of the range of a robot with respect to the lead agent, and (iv) the measurements obtained by an altimeter carried by one of the robots.

Table 1 shows a summary of the three configurations according to the kind of measurements assumed to be available in each case.

Regarding the practical scenarios where each configuration can be useful the following can be summarized: The first or second configuration can be considered for any scenario where the GPS measurements (even with low quality) are available, for instance outdoor or medium-dense environments. In particular, the second configuration will be useful in periods or circumstances where the lead agent is not seen by the monocular cameras carried by the UAVs. On the other hand, the third configuration will be more useful in any scenario or circumstance where there is no GPS availability, for instance in indoor or highly dense urban areas.

It is important to note that in a practical scenario, as it has been said before, some or all the UAVs can evolve in environments with low GPS coverage or even with no coverage. In this sense, from the implementation point of view, it is very straightforward to conceive an integrated system by simply including all the sensory inputs whenever they are available. If the minimum requirements established for each configuration are accomplished, then the integrated system will cope indistinctly with GPS-challenging or GPS-denied environments, with a performance subject to the observability criteria defined for each scenario.

## 4. Observability Analysis

As was previously mentioned, observability is a property of dynamic systems, because it has an important role in the convergence of the EKF. In this section, the observability properties of each cooperative SLAM configuration defined in Section 3 are studied. In the work of Hermann and Krener [46], it is demonstrated that a non-linear system is *locally weakly observable* if the observability rank condition rank(O)=dim(x) is verified, where O is the observability matrix.

### 4.1. Observability Matrices

In the case of the first configuration, the observability matrix $O_{1}$ can be computed from:(10)$$O_{1}={\left[\begin{array}{ccccccc} \frac{\partial (L_{f}^{0}{(}^{i}h_{c}^{j}))}{\partial x} & \frac{\partial (L_{f}^{1}{(}^{i}h_{c}^{j}))}{\partial x} & \cdots & \frac{\partial (L_{f}^{0}{(}^{h}h_{c}^{j}))}{\partial x} & \frac{\partial (L_{f}^{1}{(}^{h}h_{c}^{j}))}{\partial x} & \frac{\partial (L_{f}^{0}\left(h_{g}\right))}{\partial x} & \frac{\partial (L_{f}^{1}\left(h_{g}\right))}{\partial x} \end{array}\right]}^{T}$$
where $L_{f}^{s}h$ is the *s*-th-order Lie Derivative [47], of the scalar field h with respect to the vector field f. In (Equation 10) the zero-order and first-order Lie Derivatives are used for each measurement.

In the case of the measurements given by a monocular camera, for the projections of the landmarks, according to (Equation 3) and (Equation 1), the following zero-order Lie derivative can be defined: (11)$$\frac{\partial (L_{f}^{0}{(}^{i}h_{c}^{j}))}{\partial x}=\left[\begin{array}{cccccccccc} 0_{2\times 6} & | & 0_{2\times 6(j-1)} & {-}^{i}H_{c}^{j}{}^{W}R_{c}^{j} & 0_{2\times 3} & 0_{2\times 6(n_{2}-j)} & | & 0_{2\times 3(i-1)} & {}^{i}H_{c}^{j}{}^{W}R_{c}^{j} & 0_{2\times 3(n_{1}-i)} \end{array}\right]$$
where
(12)$${}^{i}H_{c}^{j}=\frac{f_{c}^{j}}{{{(}^{i}z_{d}^{j})}^{2}}\left[\begin{array}{ccc} \frac{{}^{i}z_{d}^{j}}{d_{u}^{j}} & 0 & -\frac{{}^{i}x_{d}^{j}}{d_{u}^{j}}\\ 0 & \frac{{}^{i}z_{d}^{j}}{d_{v}^{j}} & -\frac{{}^{i}y_{d}^{j}}{d_{v}^{j}} \end{array}\right]$$

For the same kind of measurement, the following first-order Lie Derivative can be defined: (13)$$\frac{\partial (L_{f}^{1}{(}^{i}h_{c}^{j}))}{\partial x}=\left[\begin{array}{cccccccccc} 0_{2\times 6} & | & 0_{2\times 6(j-1)} & {}^{i}H_{\mathrm{dc}}^{j} & {-}^{i}H_{c}^{j}{}^{W}R_{c}^{j} & 0_{2\times 6(n_{2}-j)} & | & 0_{2\times 3(i-1)} & {-}^{i}H_{\mathrm{dc}}^{j} & 0_{2\times 3(n_{1}-i)} \end{array}\right]$$
where
(14)$${}^{i}H_{\mathrm{dc}}^{j}=\left[\begin{array}{ccc} {}^{i}H_{1}^{j} & {}^{i}H_{2}^{j} & {}^{i}H_{3}^{j} \end{array}\right]{\left({}^{W}R_{c}^{j}\right)}^{2}v_{c}^{j}$$
and
(15)$$\begin{array}{c} {}^{i}H_{1}^{j}=\frac{f_{c}^{j}}{d_{u}^{j}{{(}^{i}z_{d}^{j})}^{2}}\left[\begin{array}{ccc} 0 & 0 & -1\\ 0 & 0 & 0 \end{array}\right]\;\;{}^{i}H_{2}^{j}=\frac{f_{c}^{j}}{d_{v}^{j}{{(}^{i}z_{d}^{j})}^{2}}\left[\begin{array}{ccc} 0 & 0 & 0\\ 0 & 0 & -1 \end{array}\right]\;\;{}^{i}H_{3}^{j}=\frac{f_{c}^{j}}{{{(}^{i}z_{d}^{j})}^{3}}\left[\begin{array}{ccc} -\frac{{}^{i}z_{d}^{j}}{d_{u}^{j}} & 0 & \frac{2^{i}x_{d}^{j}}{d_{u}^{j}}\\ 0 & -\frac{{}^{i}z_{d}^{j}}{d_{v}^{j}} & \frac{2^{i}y_{d}^{j}}{d_{v}^{j}} \end{array}\right] \end{array}$$

In the case of the measurement given by a monocular camera, for the projections of the lead agent, according to (Equation 5) and (Equation 1), the following zero-order Lie derivative can be defined: (16)$$\frac{\partial (L_{f}^{0}{(}^{h}h_{c}^{j}))}{\partial x}=\left[\begin{array}{ccccccccc} {}^{h}H_{c}^{j}{}^{W}R_{c}^{j} & 0_{2\times 3} & | & 0_{2\times 6(j-1)} & {-}^{h}H_{c}^{j}{}^{W}R_{c}^{j} & 0_{2\times 3} & 0_{2\times 6(n_{2}-j)} & | & 0_{2\times 3n_{1}} \end{array}\right]$$
where
(17)$${}^{h}H_{c}^{j}=\frac{f_{c}^{j}}{{{(}^{h}z_{d}^{j})}^{2}}\left[\begin{array}{ccc} \frac{{}^{h}z_{d}^{j}}{d_{u}^{j}} & 0 & -\frac{{}^{h}x_{d}^{j}}{d_{u}^{j}}\\ 0 & \frac{{}^{h}z_{d}^{j}}{d_{v}^{j}} & -\frac{{}^{h}y_{d}^{j}}{d_{v}^{j}} \end{array}\right]$$

For the same kind of measurement, the following first-order Lie Derivative can be defined: (18)$$\frac{\partial (L_{f}^{1}{(}^{h}h_{c}^{j}))}{\partial x}=\left[\begin{array}{ccccccccc} {-}^{h}H_{\mathrm{dc}}^{j} & {}^{h}H_{c}^{j}{}^{W}R_{c}^{j} & | & 0_{2\times 6(j-1)} & {}^{h}H_{\mathrm{dc}}^{j} & {-}^{h}H_{c}^{j}{}^{W}R_{c}^{j} & 0_{2\times 6(n_{2}-j)} & | & 0_{2\times 3n_{1}} \end{array}\right]$$
where
(19)$${}^{h}H_{\mathrm{dc}}^{j}=\left[\begin{array}{ccc} {}^{h}H_{1}^{j} & {}^{h}H_{2}^{j} & {}^{h}H_{3}^{j} \end{array}\right]{\left({}^{W}R_{c}^{j}\right)}^{2}\left(v_{c}^{j}-v_{h}\right)$$
and
(20)$$\begin{array}{c} {}^{h}H_{1}^{j}=\frac{f_{c}^{j}}{d_{u}^{j}{{(}^{h}z_{d}^{j})}^{2}}\left[\begin{array}{ccc} 0 & 0 & -1\\ 0 & 0 & 0 \end{array}\right]\;\;{}^{h}H_{2}^{j}=\frac{f_{c}^{j}}{d_{v}^{j}{{(}^{h}z_{d}^{j})}^{2}}\left[\begin{array}{ccc} 0 & 0 & 0\\ 0 & 0 & -1 \end{array}\right]\;\;{}^{h}H_{3}^{j}=\frac{f_{c}^{j}}{{{(}^{h}z_{d}^{j})}^{3}}\left[\begin{array}{ccc} -\frac{{}^{h}z_{d}^{j}}{d_{u}^{j}} & 0 & \frac{2^{h}x_{d}^{j}}{d_{u}^{j}}\\ 0 & -\frac{{}^{h}z_{d}^{j}}{d_{v}^{j}} & \frac{2^{h}y_{d}^{j}}{d_{v}^{j}} \end{array}\right] \end{array}$$

In the case of the measurement given by a GPS carried for the lead agent, according to (Equation 7) and (Equation 1), the following zero-order Lie derivative can be defined:(21)$$\frac{\partial (L_{f}^{0}\left(h_{g}\right))}{\partial x}=\left[\begin{array}{cccccc} I_{3} & 0_{3} & | & 0_{3\times 6n_{2}} & | & 0_{3\times 3n_{1}} \end{array}\right]$$
where I is the identity matrix. For the same kind of measurement, the following first-order Lie Derivative can be defined:(22)$$\frac{\partial (L_{f}^{1}\left(h_{g}\right))}{\partial x}=\left[\begin{array}{cccccc} 0_{3} & I_{3} & | & 0_{3\times 6n_{2}} & | & 0_{3\times 3n_{1}} \end{array}\right]$$

For the second configuration, the observability matrix $O_{2}$ can be computed as follows:(23)$$O_{2}={\left[\begin{array}{ccccccccc} \frac{\partial (L_{f}^{0}{(}^{i}h_{c}^{j}))}{\partial x} & \frac{\partial (L_{f}^{1}{(}^{i}h_{c}^{j}))}{\partial x} & \cdots & \frac{\partial (L_{f}^{0}{(}^{h}h_{r}^{j}))}{\partial x} & \frac{\partial (L_{f}^{1}{(}^{h}h_{r}^{j}))}{\partial x} & \frac{\partial (L_{f}^{0}\left(h_{a}^{j}\right))}{\partial x} & \frac{\partial (L_{f}^{1}\left(h_{a}^{j}\right))}{\partial x} & \frac{\partial (L_{f}^{0}\left(h_{g}\right))}{\partial x} & \frac{\partial (L_{f}^{1}\left(h_{g}\right))}{\partial x} \end{array}\right]}^{T}$$

In this case, from the measurement given by a range sensor, according to (Equation 9) and (Equation 1), the following zero-order Lie derivative can be defined:(24)$$\frac{\partial (L_{f}^{0}{(}^{h}h_{r}^{j}))}{\partial x}=\left[\begin{array}{ccccccccc} {}^{h}H_{r}^{j} & 0_{1\times 3} & | & 0_{1\times 6(j-1)} & {-}^{h}H_{r}^{j} & 0_{1\times 3} & 0_{1\times 6(n_{2}-j)} & | & 0_{1\times 3n_{1}} \end{array}\right]$$
where
(25)$${}^{h}H_{r}^{j}=\left[\begin{array}{ccc} \frac{x_{h}-x_{c}^{j}}{{}^{h}h_{r}^{j}} & \frac{y_{h}-y_{c}^{j}}{{}^{h}h_{r}^{j}} & \frac{z_{h}-z_{c}^{j}}{{}^{h}h_{r}^{j}} \end{array}\right]$$

For the same kind of measurement, the following first-order Lie Derivative can be defined:(26)$$\frac{\partial (L_{f}^{1}{(}^{h}h_{r}^{j}))}{\partial x}=\left[\begin{array}{ccccccccc} {}^{h}H_{\mathrm{dr}}^{j} & {}^{h}H_{r}^{j} & | & 0_{1\times 6(j-1)} & {-}^{h}H_{\mathrm{dr}}^{j} & {-}^{h}H_{r}^{j} & 0_{1\times 6(n_{2}-j)} & | & 0_{1\times 3n_{1}} \end{array}\right]$$
where
(27)$${\left({}^{h}H_{\mathrm{dr}}^{j}\right)}^{T}=\frac{1}{{}^{h}h_{r}^{j}}\left[I_{3}-{\left({}^{h}H_{r}^{j}\right)}^{T}{\;}^{h}H_{r}^{j}\right]\left(v_{h}-v_{c}^{j}\right)$$

In the case of the measurement given by an altimeter carried by the *j*-th quadcopter, according to (Equation 8) and (Equation 1), the following zero-order Lie derivative can be defined:(28)$$\frac{\partial (L_{f}^{0}\left(h_{a}^{j}\right))}{\partial x}=\left[\begin{array}{ccccccccc} 0_{1\times 6} & | & 0_{1\times 6(j-1)} & 0_{1\times 2} & 1 & 0_{1\times 3} & 0_{1\times 6(n_{2}-j)} & | & 0_{1\times 3n_{1}} \end{array}\right]$$

For the same kind of measurement, the following first-order Lie Derivative can be defined:(29)$$\frac{\partial (L_{f}^{1}\left(h_{a}^{j}\right))}{\partial x}=\left[\begin{array}{cccccccc} 0_{1\times 6} & | & 0_{1\times 6(j-1)} & 0_{1\times 5} & 1 & 0_{1\times 6(n_{2}-j)} & | & 0_{1\times 3n_{1}} \end{array}\right]$$

The third configuration of the system takes into account circumstances where the GPS can be unavailable. In this case, the observability matrix $O_{3}$ can be computed as follows: (30)$$O_{3}={\left[\begin{array}{ccccccccc} \frac{\partial (L_{f}^{0}{(}^{i}h_{c}^{j}))}{\partial x} & \frac{\partial (L_{f}^{1}{(}^{i}h_{c}^{j}))}{\partial x} & \cdots & \frac{\partial (L_{f}^{0}{(}^{h}h_{c}^{j}))}{\partial x} & \frac{\partial (L_{f}^{1}{(}^{h}h_{c}^{j}))}{\partial x} & \;\frac{\partial (L_{f}^{0}{(}^{h}h_{r}^{j}))}{\partial x} & \frac{\partial (L_{f}^{1}{(}^{h}h_{r}^{j}))}{\partial x} & \frac{\partial (L_{f}^{0}\left(h_{a}^{j}\right))}{\partial x} & \frac{\partial (L_{f}^{1}\left(h_{a}^{j}\right))}{\partial x} \end{array}\right]}^{T}$$

### 4.2. Theoretical Results

For the sake of presentation, a single observability matrix $O_{t}$ is expanded including all the Lie derivatives obtained from each system configuration. In this case, in order to differentiate the Lie derivatives that belong to each kind of system configuration, the rows of the matrix (31) are differentiated by colors: The rows that are common to all configurations are indicated in blue. The rows that are common to the first and second configuration are indicated in red. The rows that are common to the first and third configuration are indicated in orange. The rows that are common to the second and third configuration are indicated in black. For instance, according to the above, the observability matrix $O_{2}$ (second configuration) will be composed of selecting the blue and red rows from matrix $O_{t}$.


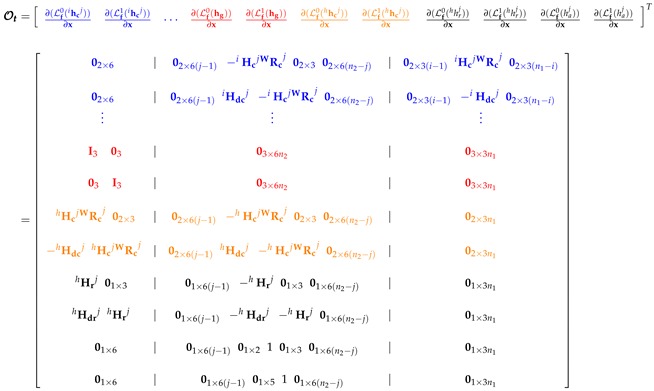
(31)

According to (Equation 10) and (31), the rank of the observability matrix $O_{1}$ is $rank\left(O_{1}\right)=3n_{1}+6n_{2}+6$, where $n_{1}$ is the number of landmarks being measured, $n_{2}$ is the number of robots and 6, is the number of states of the lead agent. In this case, $n_{1}$ is multiplied by 3, since this is the number of states per landmark given by the Euclidean parametrization, $n_{2}$ is multiplied by 6, since this is the number of states per robot given by its global position and its derivatives. Given that $dim\left(x\right)=3n_{1}+6n_{2}+6$, then, the system under its first configuration is *locally weakly observable*, because rank(O)=dim(x).

Now, according to (Equation 23) and (31), the rank of the observability matrix $O_{2}$ is $rank\left(O_{2}\right)=3n_{1}+6n_{2}+6$. Therefore, the proposed system under the second configuration is also *locally weakly observable*.

For the third configuration, according to (Equation 30) and (31), the maximum rank of the observability matrix $O_{3}$ is $rank\left(O_{3}\right)=\left(3n_{1}+6n_{2}+6\right)-2$. Therefore, $O_{3}$ will be rank deficient ($rank\left(O_{3}\right)<dim\left(x\right)$). In this case, the unobservable modes are spanned by the right nullspace basis of the observability matrix $O_{3}$, therefore:(32)$$\begin{array}{c} N=\mathrm{null}\left(O_{3}\right)=\left[\begin{array}{c} S\\ 0_{3\times 2}\\ ─\\ S\\ 0_{3\times 2}\\ \vdots \\ S\\ 0_{3\times 2}\\ ─\\ S\\ \vdots \\ S \end{array}\right],\;S=\left[\begin{array}{cc} 1 & 0\\ 0 & 1\\ 0 & 0 \end{array}\right] \end{array}$$

It is straightforward to verify that the right nullspace basis of $O_{3}$ spans for N, (i.e., $O_{3}N=0$).

From (Equation 32) it can be seen that the system is partially observable and that the unobservable modes cross with the states that correspond to the global position in *x* and *y* of the robots, the landmarks and the lead agent; these states are unobservable. An important conclusion is that all the vectors of the right null space basis are orthogonal with the rest of the states and therefore these states are completely observable.

It is very important to note, that in the absence of GPS measurements the system cannot be completely observable, given the absence of global position measurements within the system. This result is consistent with the one that can be expected from any SLAM system, which is world-centric but does not include global measurements. Therefore, in the case of the third configuration, the objective was to obtain the best possible observability result, that is, to reduce to a greater extent the number of unobservable states. It is worth noting that the first and second configurations obtain a less favorable result with respect to the third configuration in terms of the observability property when GPS measurements are not considered (see Table 2).

Some important remarks about the former observability analysis can be extracted:In the case of the first and third configurations, a single robot having visual contact with the lead agent represents a sufficient condition to obtain the previous results (see Figure 4).In the case of the second and third configurations, only a single range sensor, used for measuring the relative distance of the *j*-th quadcopter respect to the lead agent, is needed in order to obtain the previous results (see Figure 4).In the case of the second and third configurations, only a single altimeter sensor, used for measuring the altitude of the *j*-th quadcopter, is needed in order to obtain the previous results (see Figure 4).In the case of the third configuration, if two or more robots have visual contact with the lead agent, range measurements are not necessary in order to obtain the previous results.For all the configurations, in order to obtain the previous results, it is necessary to link the members of the multi-UAV system through the measurements of the landmarks (see Figure 4). In other words, a robot needs to share the observation of at least two landmarks with another robot.In the case of the third configuration, adding Lie Derivatives of higher order to the observability matrix does not improve the results.

## 5. EKF-Cooperative Monocular SLAM

The main goal of the proposed method is to estimate the system state using the EKF-based SLAM methodology. Figure 5 shows the architecture of the proposed system under its three configurations.

### 5.1. EKF-SLAM

Using (Equation 1), the following discretized system state model a can be defined:(33)$$x_{k}=f\left(x_{k-1},n_{k-1}\right)=\left[\begin{array}{c} x_{hk}\\ v_{hk}\\ x_{ck}^{j}\\ v_{ck}^{j}\\ x_{ak}^{i} \end{array}\right]=\left[\begin{array}{c} x_{hk-1}+\left(v_{hk-1}\right)\Delta t\\ \;\;\;v_{hk-1}+\zeta_{hk-1}\\ x_{ck-1}^{j}+\left(v_{ck-1}^{j}\right)\Delta t\\ \;\;\;v_{ck-1}^{j}+\eta_{ck-1}^{j}\\ x_{ak-1}^{i} \end{array}\right]$$
(34)$$n_{k}=\left[\begin{array}{c} \zeta_{hk}\\ \eta_{ck}^{j} \end{array}\right]=\left[\begin{array}{c} a_{h}\Delta t\\ \alpha_{c}^{j}\Delta t \end{array}\right]$$

For configuration 1, the system measurements are defined according to (Equation 3), (Equation 5) and (Equation 7), as: (35)$$z_{1}{}_{k}=h\left(x_{k},r_{1}{}_{k}\right)=\left[\begin{array}{c} {\;}^{i}h_{c}^{j}{}_{k}+{}^{i}r_{c}{{}_{k}}^{j}\\ {}^{h}h_{c}^{j}{}_{k}+{}^{h}r_{c}{{}_{k}}^{j}\\ \;\;h_{g}{}_{k}+r_{g}{}_{k} \end{array}\right]$$
(36)$$r_{1}{}_{k}=\left[\begin{array}{c} {}^{i}r_{c}{{}_{k}}^{j}\\ {}^{h}r_{c}{{}_{k}}^{j}\\ \;r_{g}{}_{k} \end{array}\right]$$

For configuration 2, the system measurements are defined according to (Equation 3), (Equation 7), (Equation 8) and (Equation 9), as: (37)$$z_{2}{}_{k}=h\left(x_{{}_{k}},r_{2}{}_{k}\right)=\left[\begin{array}{c} {}^{i}h_{c}^{j}{}_{k}+{}^{i}r_{c}{{}_{k}}^{j}\\ \;h_{g}{}_{k}+r_{g}{}_{k}\\ \;\;h_{ak}^{j}+r_{ak}^{j}\\ {}^{h}h_{rk}^{j}+{}^{h}r_{rk}^{j} \end{array}\right]$$
(38)$$r_{2}{}_{k}=\left[\begin{array}{c} {}^{i}r_{c}{{}_{k}}^{j}\\ \;r_{g}{}_{k}\\ \;r_{ak}^{j}\\ {}^{h}r_{rk}^{j} \end{array}\right]$$

For configuration 3, the system measurements are defined according to (Equation 3), (Equation 5), (Equation 8) and (Equation 9), as: (39)$$z_{3}{}_{k}=h\left(x_{k},r_{3}{}_{k}\right)=\left[\begin{array}{c} {}^{i}h_{c}^{j}{}_{k}+{}^{i}r_{c}{{}_{k}}^{j}\\ {}^{h}h_{c}^{j}{}_{k}+{}^{h}r_{c}{{}_{k}}^{j}\\ \;\;h_{ak}^{j}+r_{ak}^{j}\\ {}^{h}h_{rk}^{j}+{}^{h}r_{rk}^{j} \end{array}\right]$$
(40)$$r_{3}{}_{k}=\left[\begin{array}{c} {}^{i}r_{c}{{}_{k}}^{j}\\ {}^{h}r_{c}{{}_{k}}^{j}\\ \;r_{ak}^{j}\\ {}^{h}r_{rk}^{j} \end{array}\right]$$
where $a_{h}$ and $\alpha_{c}^{j}$ represent unknown linear accelerations that are assumed to have Gaussian distribution with zero mean. Moreover, let $n_{k}∼N\left(0,Q_{k}\right)$, $r_{1}{}_{k}∼N\left(0,R_{1}{}_{k}\right)$, $r_{2}{}_{k}∼N\left(0,R_{2}{}_{k}\right)$ and $r_{3}{}_{k}∼N\left(0,R_{3}{}_{k}\right)$ be the noise vectors that affect the state and the measurements, which are assumed to be mutually uncorrelated. Let Δt be the differential of time and *k* the sample step. In this work, a Gaussian random process is used for propagating the velocity of the vehicle. The proposed scheme is independent of the kind of aircraft and therefore is not restricted by the use of a specific dynamic model.

The prediction stage of the EKF is defined by:(41)$${\hat{x}}_{k}^{-}=f\left({\hat{x}}_{k-1},0\right)$$
(42)$$P_{k}^{-}=A_{k}P_{k-1}A_{k}^{T}+W_{k}Q_{k-1}W_{k}^{T}$$

The correction stage of the EKF is defined by:(43)$${\hat{x}}_{k}={\hat{x}}_{k}^{-}+K_{k}\left(z_{k}-h\left({\hat{x}}_{k}^{-},0\right)\right)$$
(44)$$P_{k}=\left(I-K_{k}C_{k}\right)P_{k}^{-}$$
with
(45)$$K_{k}=P_{k}^{-}C_{k}^{T}{\left(C_{k}P_{k}^{-}C_{k}^{T}+V_{k}R_{k}V_{k}^{T}\right)}^{-1}$$
and
(46)$$\begin{array}{c} A_{k}=\frac{\partial f}{\partial x}\left({\hat{x}}_{k-1},0\right)\;C_{k}=\frac{\partial h}{\partial x}\left({\hat{x}}_{k}^{-},0\right)\\ W_{k}=\frac{\partial f}{\partial n}\left({\hat{x}}_{k-1},0\right)\;V_{k}=\frac{\partial h}{\partial r}\left({\hat{x}}_{k}^{-},0\right) \end{array}$$
P is the covariance matrix of the system state and K is the Kalman gain.

### 5.2. Initialization of Map Features

The initialization process of new map features is carried out in a cooperative way. The 3D position of the new map features is estimated by means of a pseudo-stereo system formed by the monocular cameras mounted on a pair of UAVs that observe common landmarks. In this case, the process of initialization is carried out when a new potential landmark is observed by two cameras; if this condition is fulfilled then the landmark can be initialized by means of a linear triangulation.

The state of the new feature is computed using the *a posteriori* values obtained in the correction stage of the EKF. According to (Equation 3), the following expression can be defined in homogeneous coordinates: (47)$${}^{i}\gamma_{c}^{j}\left[\begin{array}{c} {}^{i}u_{c}^{j}\\ {}^{i}v_{c}^{j}\\ 1 \end{array}\right]=\begin{array}{cc} T_{c}^{j} & 0_{3\times 1} \end{array}{\hat{E}}_{c}^{j}\left[\begin{array}{c} x_{a}^{i}\\ 1 \end{array}\right]$$
where ${}^{i}\gamma_{c}^{j}$ is a scale factor. Additionally, it is defined:(48)$$\begin{array}{c} {\hat{E}}_{c}^{j}=\left[\begin{array}{cc} {}^{W}{\hat{R}}_{c}^{j} & {\hat{x}}_{c}^{j}\\ 0_{1\times 3} & 1 \end{array}\right]\;T_{c}^{j}=\left[\begin{array}{ccc} \frac{f_{c}^{j}}{d_{u}^{j}} & 0 & c_{u}^{j}+d_{ur}^{j}+d_{ut}^{j}\\ 0 & \frac{f_{c}^{j}}{d_{v}^{j}} & c_{v}^{j}+d_{vr}^{j}+d_{vt}^{j}\\ 0 & 0 & 1 \end{array}\right] \end{array}$$

Using (Equation 47), and considering the projection onto two any UAV cameras, a linear system can be formed in order to estimate $x_{a}^{i}$:(49)$$D^{i}x_{a}^{i}=b^{i}\;x_{a}^{i}=D^{i^{†}}b^{i}$$
where $D^{i^{†}}$ is the Moore Penrose right pseudo-inverse matrix of $D^{i}$, and
(50)$$\begin{array}{c} D^{i}=\left[\begin{array}{ccc} k_{31}^{j}{}^{i}u_{c}^{j}-k_{11}^{j} & k_{32}^{j}{}^{i}u_{c}^{j}-k_{12}^{j} & k_{33}^{j}{}^{i}u_{c}^{j}-k_{13}^{j}\\ k_{31}^{j}{}^{i}v_{c}^{j}-k_{21}^{j} & k_{32}^{j}{}^{i}v_{c}^{j}-k_{22}^{j} & k_{33}^{j}{}^{i}v_{c}^{j}-k_{23}^{j} \end{array}\right]b^{i}=\left[\begin{array}{c} k_{14}^{j}-k_{34}^{j}{}^{i}u_{c}^{j}\\ k_{24}^{j}-k_{34}^{j}{}^{i}v_{c}^{j} \end{array}\right] \end{array}$$
with
(51)$$\left[\begin{array}{cc} T_{c}^{j} & 0_{3\times 1} \end{array}\right]{\hat{E}}_{c}^{j}=\left[\begin{array}{c} k_{11}^{j}\;k_{12}^{j}\;k_{13}^{j}\;k_{14}^{j}\\ k_{21}^{j}\;k_{22}^{j}\;k_{23}^{j}\;k_{24}^{j}\\ k_{31}^{j}\;k_{32}^{j}\;k_{33}^{j}\;k_{34}^{j} \end{array}\right]$$
when a new landmark is initialized, the system state x is augmented by: $x={\left[\begin{array}{cccccc} x_{h} & v_{h} & x_{c}^{j} & v_{c}^{j} & x_{a}^{i} & x_{a}^{new} \end{array}\right]}^{T}$. And the new covariance matrix $P_{new}$ is computed by: (52)$$P_{new}=\Delta J\left[\begin{array}{cc} P & 0\\ 0 & {}^{i}R^{j} \end{array}\right]=\Delta J^{T}$$
where ΔJ is the Jacobian for the initialization function, and ${}^{i}R^{j}$ is the measurement noise covariance matrix for $\left({}^{i}u_{c}^{j}{,}^{i}v_{c}^{j}\right)$.

## 6. Control Flight Formation

This section presents the high-level control scheme that allows to carry out the formation of the UAVs with respect to the lead agent. The kinematic model is based on the leader-follower scheme presented in [48].

The kinematic model of the *j*-th UAV can be described as:(53)$$\begin{array}{c} {\dot{x}}^{j}=v_{x}^{j}\cos\left(\psi^{j}\right)-v_{y}^{j}\sin\left(\psi^{j}\right)\\ {\dot{y}}^{j}=v_{x}^{j}\sin\left(\psi^{j}\right)+v_{y}^{j}\cos\left(\psi^{j}\right)\\ {\dot{z}}^{j}=v_{z}^{j}\\ {\dot{\psi }}^{j}=\omega^{j} \end{array}$$

Let $x_{q}^{j}={\left[\begin{array}{ccc} x^{i} & y^{i} & z^{i} \end{array}\right]}^{T}$ be the position (in meters) of the *j*-th UAV, with respect to the coordinate reference system *W* (See Figure 6). Let $v_{x}^{j}$ and $v_{y}^{j}$ be the linear velocity (in m/s) components in the *x* and *y* directions of the *j*-th UAV, with respect to the coordinate reference system *Q* (located in the center of mass of the aerial vehicle). Let $v_{z}^{j}$ be the linear velocity (in m/s) component in the *z* direction of the *j*-th UAV, with respect to the coordinate reference system *W*. Let $\psi^{j}$ be the yaw angle (in radians) of the *j*-th UAV, with respect to the coordinate reference system *W*. Let $\omega^{j}$ be the angular velocity (in radians/s) for the yaw angle of the *j*-th UAV, with respect to the coordinate reference system *W*.

It is desired to maintain the *j*-th UAV to a distance $\lambda_{x}^{j}$, $\lambda_{y}^{j}$ from the lead agent (See Figure 6). Let $\lambda_{x}^{j}$, $\lambda_{y}^{j}$ be the coordinates of the lead agent with respect to the coordinate reference system *Q*. It is also desired to maintain the *j*-th UAV at an altitude differential $\lambda_{z}^{j}$ from the lead agent. Given all these considerations, the following can be defined:(54)$$\begin{array}{c} \lambda_{x}^{j}=\left(x_{h}-x^{j}\right)\cos\left(\psi^{j}\right)+\left(y_{h}-y^{j}\right)\sin\left(\psi^{j}\right)\\ \lambda_{y}^{j}=-\left(x_{h}-x^{j}\right)\sin\left(\psi^{j}\right)+\left(y_{h}-y^{j}\right)\cos\left(\psi^{j}\right)\\ \lambda_{z}^{j}=z^{j}-z_{h} \end{array}$$

Differentiating (Equation 54) with respect to time, using (Equation 1), (Equation 53) and (Equation 54), the following can be obtained:(55)$$\begin{array}{c} {\dot{\lambda }}_{x}^{j}=\omega^{j}\lambda_{y}^{j}+{\dot{x}}_{h}\cos\left(\psi^{j}\right)+{\dot{y}}_{h}\sin\left(\psi^{j}\right)-v_{x}^{j}\\ {\dot{\lambda }}_{y}^{j}=-\omega^{j}\lambda_{x}^{j}-{\dot{x}}_{h}\sin\left(\psi^{j}\right)+{\dot{y}}_{h}\cos\left(\psi^{j}\right)-v_{y}^{j}\\ {\dot{\lambda }}_{z}^{j}=v_{z}^{j}-{\dot{z}}_{h} \end{array}$$

Finally, the dynamics of the formation can be defined as follows:(56)$$\dot{\lambda }=g+u$$
where
(57)$$\begin{array}{c} \lambda =\left[\begin{array}{c} \lambda_{x}^{j}\\ \lambda_{y}^{j}\\ \lambda_{z}^{j}\\ \psi^{j} \end{array}\right]\;g=\left[\begin{array}{c} \;\;\omega^{j}\lambda_{y}^{j}+{\dot{x}}_{h}\cos\left(\psi^{j}\right)+{\dot{y}}_{h}\sin\left(\psi^{j}\right)\\ -\omega^{j}\lambda_{x}^{j}-{\dot{x}}_{h}\sin\left(\psi^{j}\right)+{\dot{y}}_{h}\cos\left(\psi^{j}\right)\\ \;\;\;\;\;\;\;\;\;\;\;\;\;\;\;\;\;\;\;\;\;-{\dot{z}}_{h}\\ \;\;\;\;\;\;\;\;\;\;\;\;\;\;\;\;\;\;\;\;\;\;\;\;0 \end{array}\right]\;u=\left[\begin{array}{c} -v_{x}^{j}\\ -v_{y}^{j}\\ \;\;v_{z}^{j}\\ \;\;\omega^{j} \end{array}\right]\; \end{array}$$

In the following, it is assumed that there exist uncertainty and disturbances coupled to the input of the system. Therefore, Equation (Equation 56) can be defined as:(58)$$\dot{\lambda }=g+\Delta_{g}+u$$
where $\Delta_{g}$ is a bounded and unknown uncertainty and disturbances term satisfying $\Delta_{g}\leq \epsilon$, being ϵ a positive constant. The control scheme is designed in order to allow that the proposed dynamics of the formation converges to the desired values. In this case, the sliding mode control technique [49] is proposed to be used for the development of a robust controller. The state values required by the control system are obtained from the estimator described in Section 5. The yaw angle of the *j*-th UAV is obtained through an AHRS onboard the aerial vehicle. To obtain the control laws by means of an analysis in continuous time it is assumed that the estimated value is passed through a zero order hold (ZOH) (See Figure 7).

Since the estimated state of the position of the UAV is defined with respect to the reference system C (see Section 2 and Section 5), it is necessary to apply a transformation to the estimated state ${\hat{x}}_{c}^{j}$ for obtaining ${\hat{x}}_{q}^{j}$, which in turn, is necessary to obtain the control laws. Therefore, the following equation is defined:(59)$${\hat{x}}_{q}^{j}={\hat{x}}_{c}^{j}-{}^{q}d_{c}^{j}$$
where ${}^{q}d_{c}^{j}$ is the translation vector (in meters) from the coordinate reference system *Q* to the coordinate reference system *C*. Please note that ${}^{q}d_{c}^{j}$ is assumed to be known and constant.

Firstly, the next equation is defined:(60)$$s_{\lambda }=e_{\lambda }+K_{1}\int_{0}^{t}e_{\lambda }dt$$
where $e_{\lambda }=\hat{\lambda }-\lambda_{d}$, and $\lambda_{d}$ are the desired values and $K_{1}$ is a positive definite diagonal matrix of adequate dimensions. Then, deriving (Equation 60) and substituting the dynamics defined in (Equation 58) as well as taking u as the control input, the following equation can be obtained:(61)$${\dot{s}}_{\lambda }=-{\dot{\lambda }}_{d}+K_{1}e_{\lambda }+\hat{g}+\Delta_{g}+u$$

The following control law is proposed:(62)$$u=+{\dot{\lambda }}_{d}-K_{1}e_{\lambda }-\hat{g}-K_{2}\mathrm{sign}\left(s_{\lambda }\right)$$
where $K_{2}$ is a positive definite diagonal matrix of adequate dimensions.

To prove the stability of the flight formation system dynamics, the following Lyapunov candidate function is proposed:(63)$$V_{\lambda }=\frac{1}{2}s_{\lambda }^{T}s_{\lambda }$$
with derivative
(64)$$\begin{array}{c} {\dot{V}}_{\lambda }=s_{\lambda }^{T}{\dot{s}}_{\lambda }=s_{\lambda }^{T}\left(-{\dot{\lambda }}_{d}+K_{1}e_{\lambda }+\hat{g}+\Delta_{g}+u\right) \end{array}$$
substituting (Equation 62) in (Equation 64), it is obtained:(65)$$\begin{array}{c} {\dot{V}}_{\lambda }=s_{\lambda }^{T}\left(\Delta_{g}-K_{2}\mathrm{sign}\left(s_{\lambda }\right)\right)\leq ∥s_{\lambda }∥\epsilon -∥s_{\lambda }∥\alpha \leq ∥s_{\lambda }∥\left(\epsilon -\alpha \right) \end{array}$$
where $\alpha =\lambda_{min}\left(K_{2}\right)$. So if α is chosen such that α>ϵ, it is assured that ${\dot{V}}_{\lambda }$ is negative definite. Therefore, the formation system dynamics will reach and remain in the surface $s_{\lambda }=0$ in a finite time.

## 7. Computer Simulations Results

### 7.1. Simulation Setup

The performance of the proposed navigation architecture has been assessed by means of numerical simulations. To this aim, a simulation environment has been developed, The environment is composed of 3D landmarks, randomly distributed over the ground (See Figure 8). To execute the tests, two Quadcopters, equipped with the set of sensors required by the proposed method, are emulated to follow (flying in formation) a lead agent moving freely in the environment. For the first set of tests, only the estimations obtained from the SLAM system will be evaluated, and thus, it will assumed that a control system exists capable of maintaining a perfect flying formation of the aerial robots with respect to the lead agent. For the second set of tests, the system state estimated by the SLAM will be used as feedback to the control scheme proposed in Section 6, in order to evaluate the closed-loop performance of the system.

In computer simulations, it is assumed that the initial condition of the UAVs and the lead agent states are known with certainty. The measurements from the sensors are emulated to be taken with a frequency of 10 Hz. The intrinsic parameters used for the cameras are $f_{c}^{j}/d_{u}^{j}=f_{c}^{j}/d_{v}^{j}=200.1$ and $c_{u}^{j}=c_{v}^{j}=500$. In simulations, it is also assumed that there exist enough landmarks in the environment that allow a subset of them to be observed in common by the cameras of the UAVs.

To emulate the system uncertainty, the following Gaussian noise is added to measurements: Gaussian noise with $\sigma_{c}=3$ pixels is added to the measurements given by the cameras. Gaussian noise with $\sigma_{a}=50$ cm is added to the measurements given by the altimeter and Gaussian noise with $\sigma_{r}=50$ cm is added to the measurements given by the range sensor. Gaussian noise with $\sigma_{h}=0.05$ radians is added to the measurements given by the AHRS system. In the case of GPS measurements, a Gaussian noise with $\sigma_{g}=1.5$ m is added. It is important to note that the noise considered for emulating GPS measurements is approximately three times bigger than the typical magnitude of the noise of a real GPS. In this case, since a local coordinate reference system is used, instead of a Geo-referenced one, then the most important source of error of the GPS that affects the local estimations is the random noise. The reason for emulating such extremely poor GPS performance is for showing that the proposed system is robust enough for dealing with very uncertain position data.

In practical applications, there are several factors that affect and degrade the performance of the Visual SLAM systems, the problem of data association is one of the main problems that appear when implementing the Visual SLAM algorithms. In addition, in cooperative visual systems, the data association problem is extended from the single-image case to the multiple-image case. Also, the gimbals used for stabilizing the cameras can be subject to small errors in its operation.

To take into account the above practical considerations, the following aspects are also considered in computer simulations: (i) outliers for the visual data association in each camera; (ii) outliers for the cooperative visual data association; (iii) perturbations in the orientation of the cameras. To emulate the failures of the visual data association process, 5% of the total number of visual correspondences are forced to be outliers in a random manner. In this case, each outlier is modeled by means of a measurement error $e_{m}=\sqrt{e_{u}^{2}+e_{v}^{2}}$ pixels. In this case, $e_{u}$ and $e_{v}$ are the errors in the coordinate *u* and *v* (in pixels) of the projection of the landmark over the image of the camera, given by the false value in the correspondence with respect to the real value in the correct correspondence, and $e_{m}\in \left[0,15\right]$ with a continuous uniform distribution. The errors in cameras orientation, due to the gimbal assumption, are emulated by perturbation of the actual orientation by means of the following sinusoidal function $e_{r}=0.04\sin\left(t\cdot 0.3\right)$ radians. Table 3 shows the number of failures introduced into the simulation due to the data association problem.

### 7.2. SLAM Simulation Results

In the tests presented in this section, only the estimation problem is addressed, and thus, it is assumed that a control system exists capable of maintaining the flying formation of the aerial robots with respect to the lead agent.

In this case, the trajectories followed for the Quad 1, the Quad 2 and the lead agent are given by the following parametric function:(66)$$\begin{array}{c} x_{c}^{j}\left(t\right)=x_{h}\left(t\right)=\frac{100\cos(t\cdot 0.015)}{1+\sin^{2}\left(t\cdot 0.015\right)}\\ y_{c}^{j}\left(t\right)=y_{h}\left(t\right)=\frac{100\sin(t\cdot 0.015)\cos(t\cdot 0.015)}{1+\sin^{2}\left(t\cdot 0.015\right)}\\ z_{c}^{j}\left(t\right)=z_{h}\left(t\right)=2\sin\left(t\cdot 0.03\right) \end{array}$$
with the following initial conditions: $x_{c}^{1}={\left[\begin{array}{ccc} -1.5 & 0 & 15 \end{array}\right]}^{T}$, $x_{c}^{2}={\left[\begin{array}{ccc} 1.5 & 0 & 17 \end{array}\right]}^{T}$ and $x_{h}={\left[\begin{array}{ccc} 0 & 0 & 0 \end{array}\right]}^{T}$. Figure 8 shows the trajectories followed by the lead agent, the Quad 1 and the Quad 2.

#### 7.2.1. Comparative Study

A comparative study has been carried out in order to have an insight into the performance of the proposed system under its three configurations. In this case, the comparison is carried out with respect to a single-UAV SLAM system. Therefore, the study also allows to observe the advantages and drawbacks of multi-UAV systems compared with single robot systems. The computer simulation setup for the three proposed cooperative SLAM configurations, both for the case of multi-robot and single robot are described below:For the first system configuration, the features initialization process is carried out as shown in Section 5, having Quad 1 visual contact with the lead agent. In the case of the single-UAV system, only one quadcopter (Quad 1) is used for obtaining measurements instead of two. In this latter case, the features initialization process is carried out as shown in [50]. In both systems, the GPS is carried by the Quad 1.For the second system configuration, the simulation setup is similar to the one used for testing the first configuration, but in this case, the lead agent is maintained outside the field of view of the robots. In addition, range measurements are obtained from the Quad 1 with respect to the lead agent, and an altimeter is carried by the Quad 1.For the third system configuration, the simulation setup is similar to the one used for testing the second system configuration, but in this case, no GPS measurements are considered. Instead, visual measurements of the lead agent obtained from the Quad 1 are considered.

Figure 9 shows the real and estimated trajectory obtained from the Multi-UAV system and the single-UAV system. For better comparison purposes note that the flying trajectory is divided into three stages: (i) during the time 0 to 70 s, the system is tested under its first configuration, (ii) during the time 70 to 140 s, the system is tested under its second configuration, and (iii) during the time 140 to 210 s, the system is tested in under the third configuration. Please note that, in this case, for the sake of clarity, for the case of the proposed Multi-UAV system, only the estimated trajectory of the Quad 1 is presented. The results of the estimated state of the Quad 2 are very closely similar to those presented for the Quad 1.

Figure 10 shows the evolution over time of the absolute value of the error for the estimated trajectory of the lead agent ($e^{h}$) and the Quad 1 ($e^{1}$) with respect to the global reference system *W* (i.e., $e_{x}^{h}=x_{h}-{\hat{x}}_{h}$, $e_{y}^{h}=y_{h}-{\hat{y}}_{h}$, $e_{z}^{h}=z_{h}-{\hat{z}}_{h}$, $e_{x}^{1}=x_{c}^{1}-{\hat{x}}_{c}^{1}$, $e_{y}^{1}=y_{c}^{1}-{\hat{y}}_{c}^{1}$ and $e_{z}^{1}=z_{c}^{1}-{\hat{z}}_{c}^{1}$.

Table 4 summarizes the Mean Squared Error (MSE) for the estimated position in the three axes of the lead agent and of the Quad 1. In this case, Table 4 shows the MSE obtained during each system configuration and the MSE obtained for the whole trajectory. According to the above results, it can be seen that the Multi-UAV system presents a better performance. Table 5 provides an insight into the performance of the proposed method for estimating the location of the 3D landmarks composing the environment. In this case, the total (sum of all) of the Mean Squared Errors for the estimated position of the landmarks is presented. Also, the total of the Mean Squared Errors for the initial estimated position of the landmarks is presented. Please note that the results are presented for each coordinate of the reference frame *W*. Please note that Table 4 and Table 5 show the errors obtained during each system configuration as well as the errors obtained for the whole trajectory. The results show that the proposed Multi-UAV method has a much better performance than the single-UAV system, regarding the error obtained in the estimation of landmarks position.

To evaluate the statistical consistency of each method, the average NEES (Normalized Estimation Error Squared [51]) over $n_{3}$ Monte Carlo runs was computed, as it is proposed in [52]. The NEES is estimated as follows:(67)$$\epsilon_{k}=\left[x_{k}-{\hat{x}}_{k}\right]P_{k}^{-1}\left[x_{k}-{\hat{x}}_{k}\right]$$

In addition, the average NEES is computed from:(68)$${\bar{\epsilon }}_{k}=\frac{1}{n_{3}}\sum_{r=1}^{n_{3}}\epsilon_{k}^{r}$$

Figure 11 shows the average NEES over 50 Monte Carlo runs obtained for each system (multi-UAV and Single-UAV). The average NEES is calculated taking into account the eighteen variables that define the complete state of the UAVs and the lead agent (position and linear velocity). It is very interesting to note how the consistency of the filter considerably degenerates in the case of the single-robot system under its third configuration, this effect happens when the system is not fully observable. On the other hand, for the multi-UAV system case, the consistency of the filter remains practically stable. Given the previous results, the proposed Multi-UAV system presents a good performance in its three configurations, despite all the failures and disturbances introduced into the system. The above study provides a good insight into the robustness of the proposed system.

#### 7.2.2. GPS-Denied Test

A set of simulations tests were carried out with the intention of evaluating the performance of each system configuration under GPS-denied environments. For this test, the flying trajectory is divided into two stages: (i) during time period 0 to 105 s, the vehicles move with GPS availability, (ii) during time period 105 to 210 s, the vehicles move with without GPS availability. Throughout the trajectory, the state of the system is estimated independently using each system configuration.

The following aspects are considered in this test: For configuration 1, altimeter measurements are also included, since, in practice, almost every UAV is equipped with this kind of sensor. For the configuration 3, during the first stage of the trajectory, GPS measurements are also included. These previous considerations will allow evaluating the performance of configuration 3 both in GPS-available environments and in GPS-denied environments.

Figure 12 shows for each configuration the evolution of the error over the time for the estimated position of the Quad 1 and the lead agent too, (i.e., $e_{r}^{h}=\sqrt{{\left(x_{h}-{\hat{x}}_{h}\right)}^{2}+{\left(y_{h}-{\hat{y}}_{h}\right)}^{2}+{\left(z_{h}-{\hat{z}}_{h}\right)}^{2}}$ and $e_{r}^{1}=\sqrt{{\left(x_{c}^{1}-{\hat{x}}_{c}^{1}\right)}^{2}+{\left(y_{c}^{1}-{\hat{y}}_{c}^{1}\right)}^{2}+{\left(z_{c}^{1}-{\hat{z}}_{c}^{1}\right)}^{2}}$). Given the above results, it can be observed that the three configurations yield good results in GPS-available environments, and it is also confirmed that configuration 3 provides good results in GPS-denied environments. However, the performance of the configuration 1 and 2 clearly degenerates when there is no GPS availability. Although configuration 1 uses at least two monocular measurements of the lead agent allowing to obtain three-dimensional information from the lead agent, is interesting to note that its performance is not so good in GPS-denied environments as expected. On the other hand, the better three-dimensional information is obtained by the sensory fusion of a monocular measurement and a range measurement with respect to the lead agent (as in configuration 3). The above results are produced by the fact that the 3D reconstruction from monocular measurements is more sensitive to noise.

### 7.3. Control Simulations Results

A set of simulations was also carried out, where the SLAM estimates are used as feedback to the control laws involved in commanding the formation of the UAVs with respect to the lead agent (see Section 6). This will allow observing the performance of both the estimation and control in a closed-loop manner.

Based on the same simulation setup used previously, now, the trajectory followed by the UAVs is given by a kinematic model with control inputs. The initial conditions of the position of the quadcopters are the same used previously. In the case of the yaw angle of the aerial vehicles, the initial conditions are as follows: $\psi^{1}=1$ and $\psi^{2}=\frac{1}{2}$. The vector $\lambda_{d}$, that defines the desired values for the flight formation is: $\lambda_{d}^{1}={\left[\begin{array}{cccc} 2.5, & 1, & 14, & \arctan(\frac{{\hat{y}}_{h}}{{\hat{x}}_{h}}) \end{array}\right]}^{T}$ and $\lambda_{d}^{2}={\left[\begin{array}{cccc} 0.5, & -1, & 16, & \arctan(\frac{{\hat{y}}_{h}}{{\hat{x}}_{h}}) \end{array}\right]}^{T}$.

In this test, the trajectory is divided into three stages, each one for testing each system configuration. Figure 13 shows the trajectories of the three elements composing the flight formation.

Figure 14 shows the evolution of the error with respect to the desired values $\lambda_{d}$ which define the flight formation.

Figure 15 shows, for each configuration, the evolution of the error over the time for the estimated position of the Quad 1 as well as the lead agent, when these estimates are used as feedback to the control laws.

Given the previous results, it can be observed that the estimation with the three configurations and the control have good performance in closed-loop.

## 8. Conclusions

In this work, the problem of the cooperative visual-based SLAM for the class of multi-UAV systems that integrate a lead agent has been addressed. For this purpose, three system configurations were proposed and investigated by means of an intensive nonlinear observability analysis. The main differences between the system configurations have to do with the set of sensors used in each case, additional to the monocular cameras carried by each aerial vehicle, as well as the circumstances they operate in a suitable manner. The objective of each configuration is to ensure that the system can be able to maximize the property of observability under different scenarios. In this case, several theoretical results were obtained from the analysis; for instance, sufficient conditions required for obtaining the observability results were established.

The first or second configuration can be considered for any scenario where the GPS measurements (even low quality measurements) are available. In particular, the second configuration will be useful in circumstances where the lead agent is not seen by cameras carried by the UAVs. The third configuration will be more useful in any scenario or circumstance where there is no GPS availability. The proposed structure based on different system configurations facilitates the mathematical and experimental analysis. However, from the implementation point of view, it will be more convenient to use an integrated system which includes all the sensory inputs whenever they are available.

The proposed system configurations were implemented using the standard EKF-based SLAM methodology. In this case, the initialization process of new map features is carried out in a cooperative way. In addition, a high-level control scheme was proposed allowing the control of the UAVs formation with respect to the lead agent.

An extensive set of computer simulations was performed in order to validate the proposed scheme, for instance: (i) the proposed multi-UAV system was compared against a single-UAV system, (ii) the consistency of the filter was investigated by means of the average NEES test, (iii) the performance of each system configuration were evaluated under GPS-denied environments, and (iv) the performance of both the estimation and the control were evaluated by analyzing the behavior of each system configuration in closed-loop.

In the above cases, several aspects regarding applicability in real scenarios of the proposed approach were considered, for instance, the association problem of visual data for each camera and the pseudo-stereo matching as well, and also the errors induced by the assumption of the camera gimbal were emulated in simulations.

Based on the results of the simulations, it can be observed how the proposed method improves the estimation of the state by considering the multi-robot cooperative scheme. Also, it is shown that the estimates produced by the proposed SLAM system can be used directly as feedback to the high-level control laws that command the flight formation. The simulation results also show that the proposed scheme is able to offer a good performance, even in the absence of GPS measurements.

However, although computer simulations are useful for evaluating the full statistical consistency of the methods, they can still neglect important practical issues that appear when the methods are used in real scenarios. In this sense, it is important to note that future work could be focused on developing experiments with real data in order to fully evaluate the applicability of the proposed approach.

## Figures and Tables

**Figure 1 sensors-18-04243-f001:**
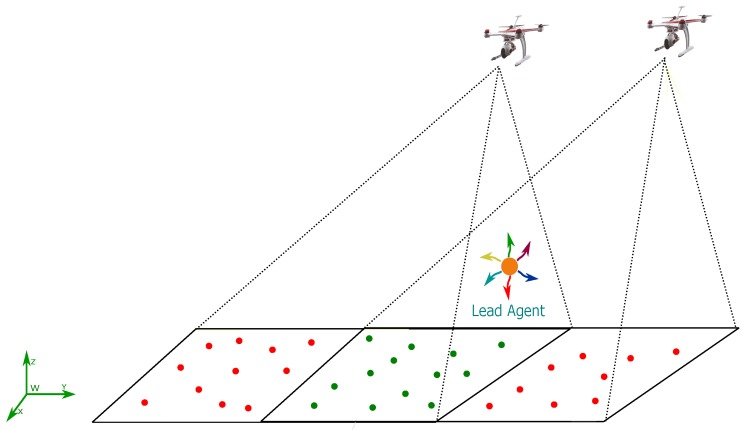
Multi-UAV SLAM system with a dynamic lead agent.

**Figure 2 sensors-18-04243-f002:**
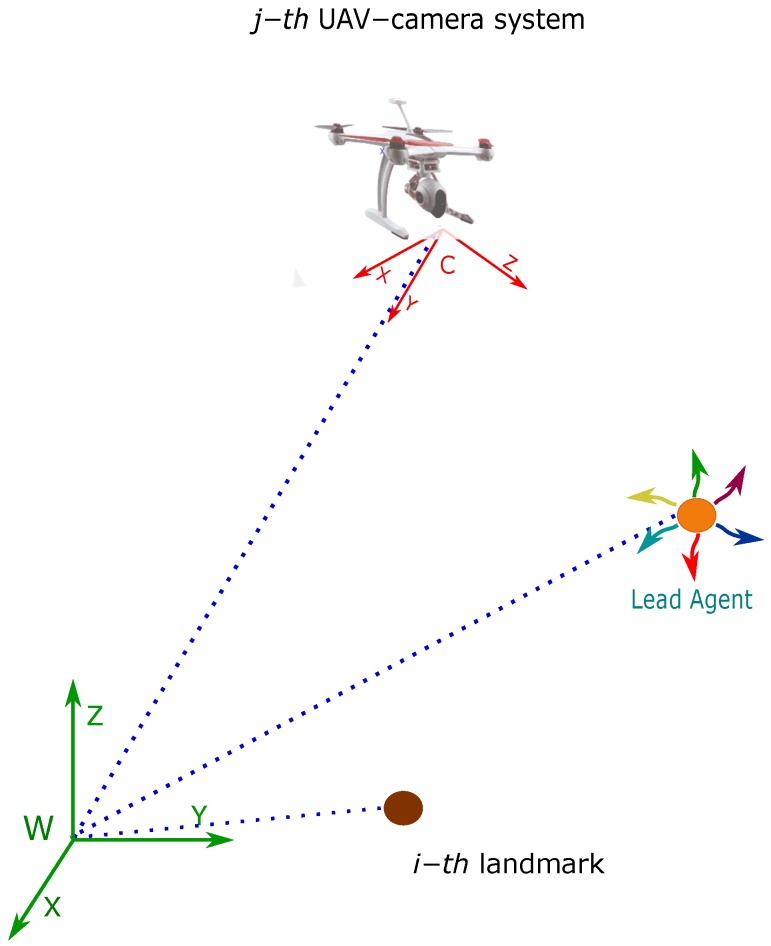
Coordinate reference systems.

**Figure 3 sensors-18-04243-f003:**
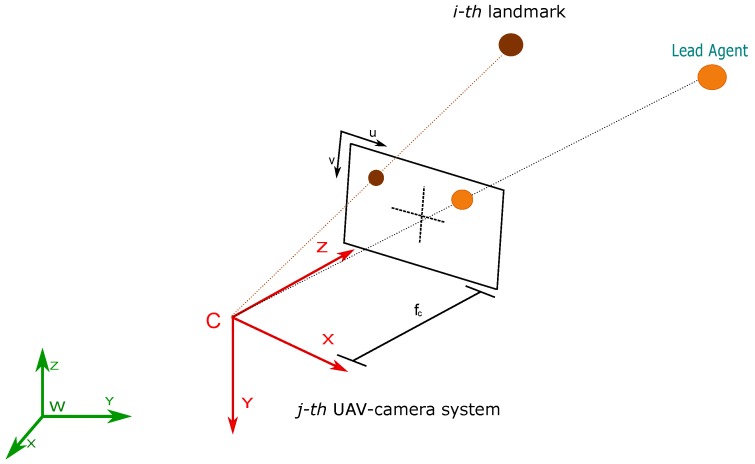
Pinhole camera projection model.

**Figure 4 sensors-18-04243-f004:**
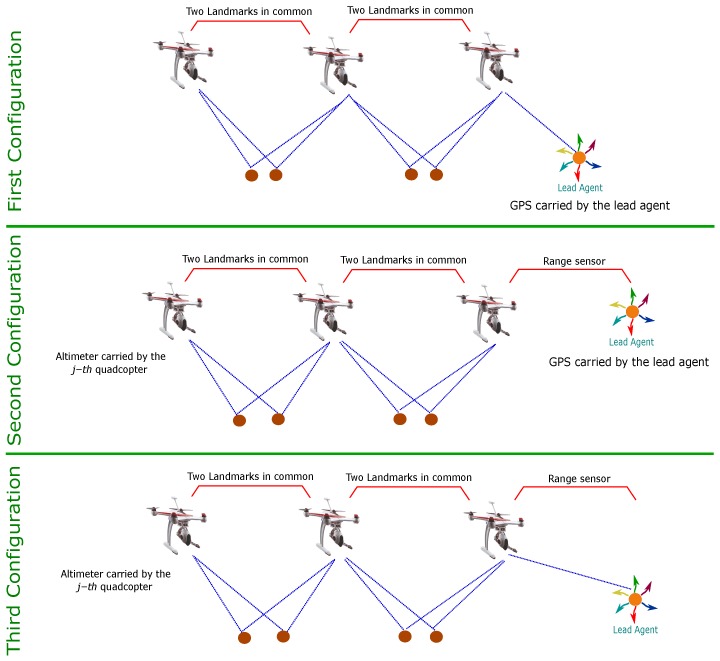
System configurations and minimum requirements for obtaining the results of the observability analysis.

**Figure 5 sensors-18-04243-f005:**
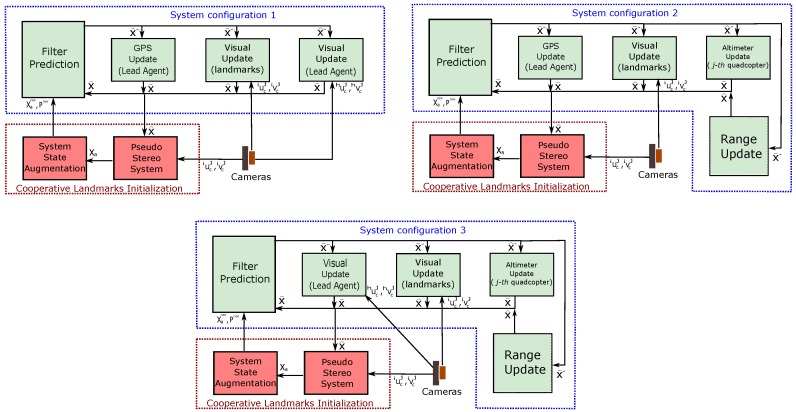
EKF-Cooperative SLAM: Block diagram showing the architecture of the system under its three configurations.

**Figure 6 sensors-18-04243-f006:**
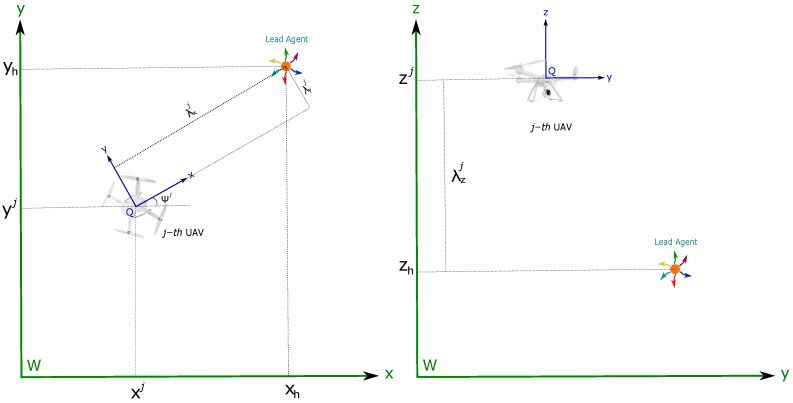
UAVs-lead agent flight formation.

**Figure 7 sensors-18-04243-f007:**
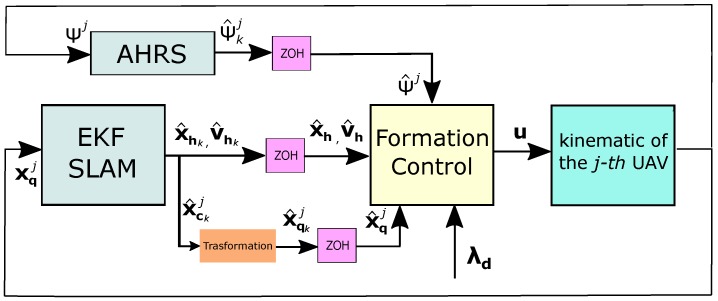
Control scheme.

**Figure 8 sensors-18-04243-f008:**
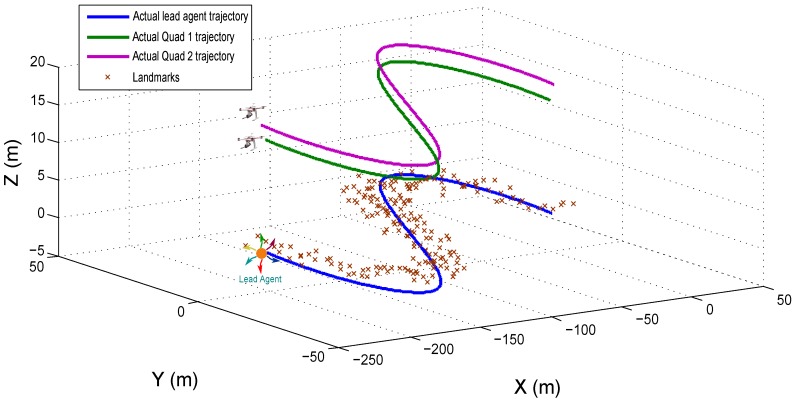
Simulation environment and trajectories followed by the lead agent, the Quad 1 and the Quad 2.

**Figure 9 sensors-18-04243-f009:**
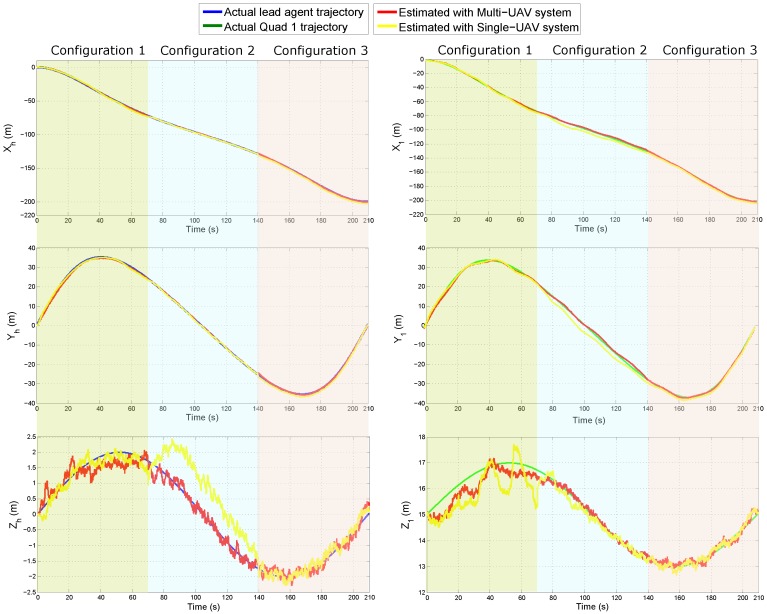
Estimated state of the lead agent (**left plots**) and of the Quad 1 (**right plots**), for both kind of system, under the three configurations.

**Figure 10 sensors-18-04243-f010:**
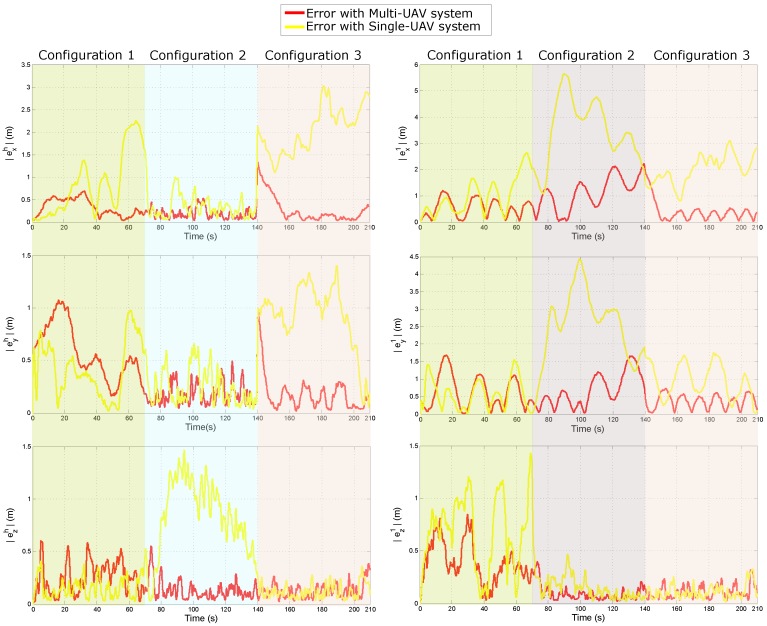
Comparison of the quality of the estimated trajectory obtained with the Multi-UAV system respect to the trajectory obtained with the Single-UAV system. **Left column** of plots illustrates the errors obtained for the lead agent. **Right column** of plots illustrates the errors obtained for the Quad 1.

**Figure 11 sensors-18-04243-f011:**
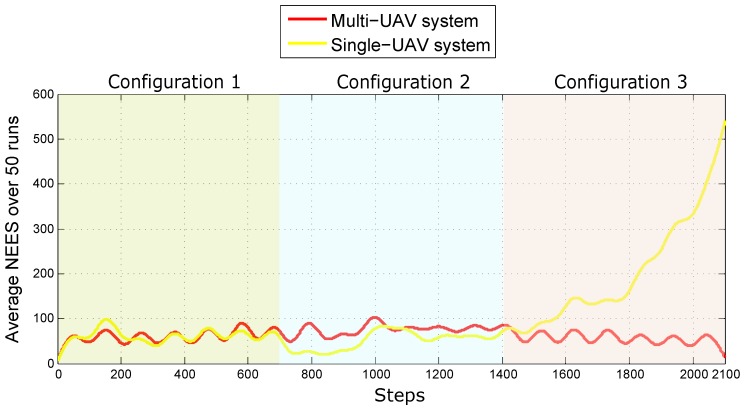
Average NEES obtained with both systems.

**Figure 12 sensors-18-04243-f012:**
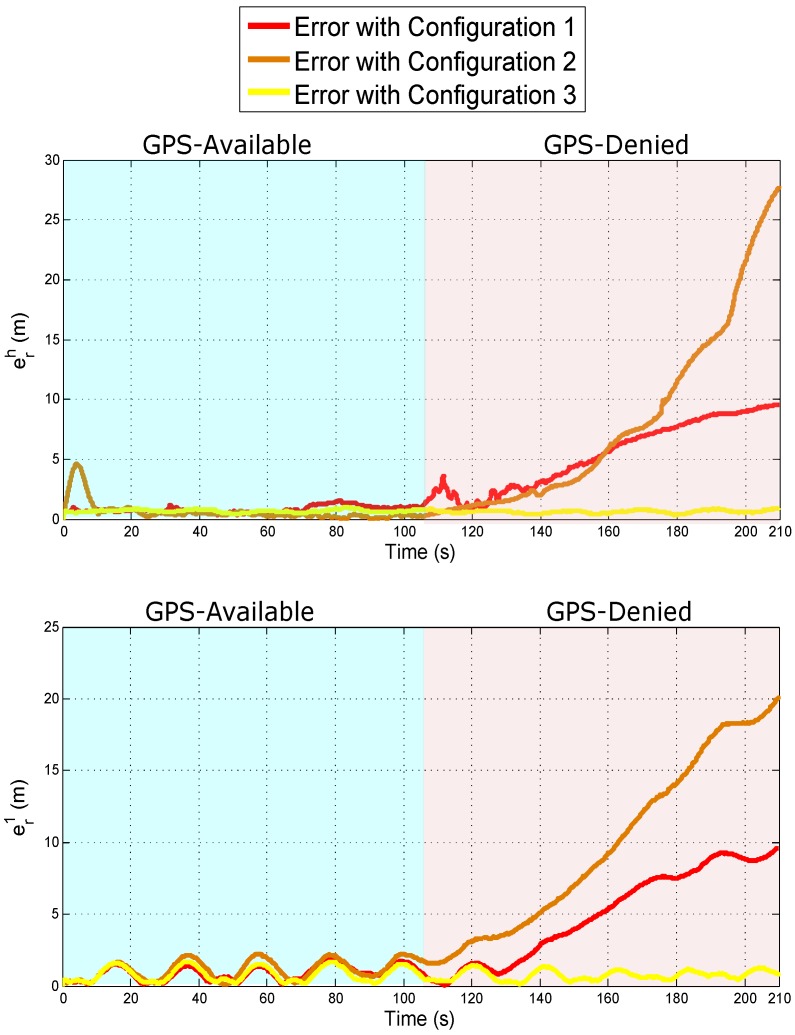
Comparison of the quality in the estimation of the position of the lead agent (**upper plot**) and the Quad 1 (**bottom plot**) under GPS-denied environments.

**Figure 13 sensors-18-04243-f013:**
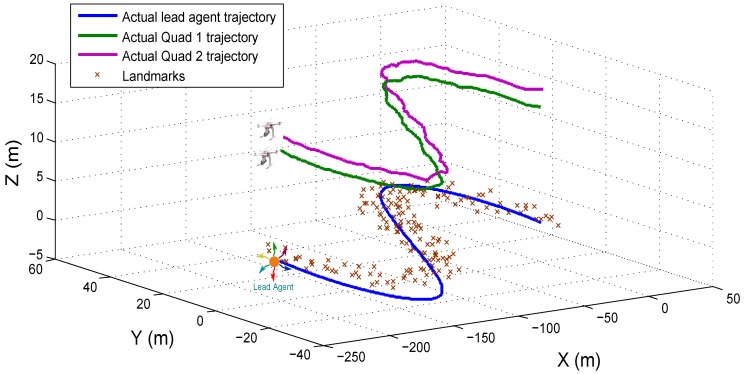
Trajectories followed by the lead agent, the Quad 1 and the Quad 2 with the system in closed-loop.

**Figure 14 sensors-18-04243-f014:**
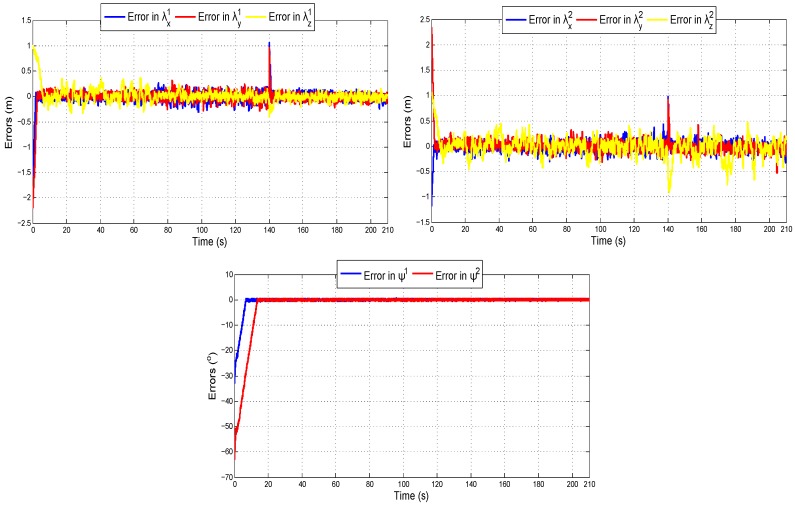
Errors in $\lambda_{d}$ during the flight trajectories.

**Figure 15 sensors-18-04243-f015:**
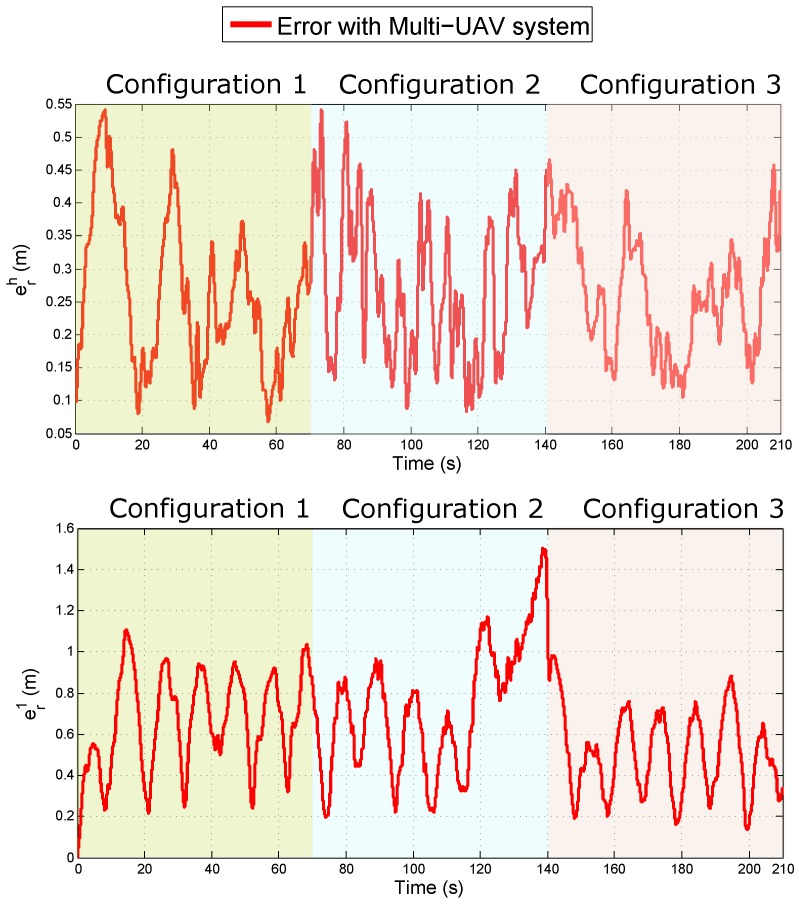
Comparison of the quality of the estimated position of the lead agent (**upper plot**) and the Quad 1 (**bottom plot**) in closed-loop.

**Table 1 sensors-18-04243-t001:** System configurations summary.

Configuration	Monocular Measurements of Landmarks	Monocular Measurements of Lead Agent	Altimeter on UAV	Range to Lead Agent	GPS on Lead Agent
First	✓	✓	✗	✗	✓
Second	✓	✗	✓	✓	✓
Third	✓	✓	✓	✓	✗

**Table 2 sensors-18-04243-t002:** Results of observability in the absence of GPS.

	Unobservable	Unobservable	Observable
	Modes	States	States
Configuration 1	4	x	-
Configuration 2	6	$x_{h}$, $v_{h}$, $x_{c}^{j}$, $y_{c}^{j}$, $x_{a}^{i}$, $y_{a}^{i}$	$z_{c}^{j}$, $z_{a}^{i}$, $v_{c}^{j}$
Configuration 3	2	$x_{h}$, $y_{h}$, $x_{c}^{j}$, $y_{c}^{j}$, $x_{a}^{i}$, $y_{a}^{i}$	$z_{h}$, $z_{c}^{j}$, $z_{a}^{i}$, $v_{h}$, $v_{c}^{j}$

**Table 3 sensors-18-04243-t003:** Number of failures introduced into the simulation.

	Visual Outliers	Visual Outliers	Visual Outliers
	(Quad 1)	(Quad 2)	(Cooperative)
Multi-UAV system	9002	8400	1706
Single-UAV system	9535	-	-

**Table 4 sensors-18-04243-t004:** Mean Squared Error for the estimated position of the lead agent ($MSEX_{h}$, $MSEY_{h}$, $MSEZ_{h}$) and the estimated position of Quad 1 ($MSEX_{1}$, $MSEY_{1}$, $MSEZ_{1}$).

		${\mathrm{MSEX}}_{h}\left(m\right)$	${\mathrm{MSEY}}_{h}\left(m\right)$	${\mathrm{MSEZ}}_{h}\left(m\right)$	${\mathrm{MSEX}}_{1}\left(m\right)$	${\mathrm{MSEY}}_{1}\left(m\right)$	${\mathrm{MSEZ}}_{1}\left(m\right)$
**Config. 1**	**Multi-UAV system**	0.1454	0.3943	0.0873	0.4063	0.6115	0.1923
	**Single-UAV system**	1.1873	0.2456	0.0362	1.4233	0.5061	0.5849
**Config. 2**	**Multi-UAV system**	0.0538	0.0457	0.0292	1.5706	0.6709	0.0163
	**Single-UAV system**	0.2047	0.0935	0.8267	13.8187	7.3878	0.0409
**Config. 3**	**Multi-UAV system**	0.1670	0.0627	0.0241	0.3093	0.1718	0.0180
	**Single-UAV system**	4.4490	0.9560	0.0231	4.3048	1.1514	0.0184
**Total**	**Multi-UAV system**	0.1221	0.1676	0.0468	0.7621	0.4847	0.0755
	**Single-UAV system**	1.9472	0.4317	0.2953	6.5140	3.0151	0.2148

**Table 5 sensors-18-04243-t005:** Total Mean Squared Error in the position estimation of the landmarks ($MSEX_{a}$, $MSEY_{a}$, $MSEZ_{a}$). Total Mean Squared Error in the initial position estimation of the landmarks ($MSEX_{ai}$, $MSEY_{ai}$, $MSEZ_{ai}$).

		${\mathrm{MSEX}}_{a}\left(m\right)$	${\mathrm{MSEY}}_{a}\left(m\right)$	${\mathrm{MSEZ}}_{a}\left(m\right)$	${\mathrm{MSEX}}_{\mathrm{ai}}\left(m\right)$	${\mathrm{MSEY}}_{\mathrm{ai}}\left(m\right)$	${\mathrm{MSEZ}}_{\mathrm{ai}}\left(m\right)$
**Config. 1**	**Multi-UAV system**	0.2794	0.4699	1.1571	0.9293	1.1608	4.3764
	**Single-UAV system**	2.5504	2.1782	18.5759	10.1012	7.9064	57.0486
**Config. 2**	**Multi-UAV system**	0.7379	0.4515	1.5301	2.0737	1.6544	8.8474
	**Single-UAV system**	1.2181	2.5109	13.2295	5.0821	9.0028	47.9993
**Config. 3**	**Multi-UAV system**	0.4126	0.4884	0.8722	0.8332	0.8006	1.5070
	**Single-UAV system**	1.5092	1.2707	2.6635	3.2802	3.7381	16.2915
**Total**	**Multi-UAV system**	0.4702	0.4683	1.2120	1.2452	1.2080	4.9143
	**Single-UAV system**	1.8385	2.0462	12.6057	6.5731	6.9910	42.2073

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
