# Peer review of "Visual-Based SLAM Configurations for Cooperative Multi-UAV Systems with a Lead Agent: An Observability-Based Approach"

_sensors, 2018, doi:10.3390/s18124243_

Reviewer 1 Report

·       Lines 4-6. Please clarify what do you mean by target. Is it the “lead agent”? I suggest unifying the adopted terminology to avoid confusions. Also, the authors should clarify the “leading role” of the lead agent in the cooperative swarm.

·       Introduction. The authors should better clarify the scenario under analysis by

o   clarifying the leading role of the lead agent;

o   clarifying whether all the UAVs are flying in GNSS-challenging or denied environment;

o   mentioning practical scenarios for which each of the proposed configuration can be useful.

·       Related work. Lines 47-50. I do not get the point of this statement. In this reviewer’s opinion, it is unlikely that, in general, purely vision-based SLAM approaches for applications involving UAVs perform better than visual inertial methods. Most of the main works in the literature regarding vision-based SLAM on board UAVs take also advantage of the inertial sensors which will always be available on board. Also, innovation in low-cost MEMS technologies is leading to more accurate sensors.

·       Related work. Lines 67-70. Please, could you better clarify the conclusion of this discussion. How is this statement related to the overview of previous works?

·       Objectives and contributions. Line 75. Please define “complementary sensory inputs”.

·       Objectives and contributions. Please can you clarify the new contribution with respect to your recent paper

Trujillo, J. C., Munguia, R., Guerra, E., & Grau, A. (2018). Cooperative Monocular-Based SLAM for Multi-UAV Systems in GPS-Denied Environments. Sensors (Basel, Switzerland), 18(5).

·       Section 2. Please can you clarify what do you mean by landmarks? Are they cooperative known objects located in the unknown environment or natural features extracted by the visual cameras?

·       Section 2. Line 140-142. Please clarify the meaning of this assumption with regards to practical applications of the proposed method.

·       Section 2. Line 144-146. A critical problem that the authors are not addressing is the detection and tracking of the lead agent by the visual cameras onboard the UAV. Do you foresee to address this problem taking advantage of the cooperative nature of the formation or in non-cooperative way? Indeed, many recent works are addressing these problems. It would be good to mention all the assumptions adopted in this work in the introduction with proper references to the solutions proposed in the literature (for instance for the problem of vision-based detection and tracking of cooperative UAVs).

·       Section 2. Lines 150-151. Again, the geometrical configuration of the formation shall be clarified, also to explain which the practical scenarios under analysis can be. You are assuming that the lead agent is located below the other UAVs, but it is the only one having the GPS. Can you clarify this assumption?

·       Lines 290-292. I have to disagree. Standalone GPS position fix has an accuracy of meters order. So the assumption is not conservative as it is stated.

·       Overall, the proposed methodology and the analysis carried out in this manuscript are indeed interesting. The main problem that the authors should address is to better clarify the potential scenario of interest for each of the proposed configurations to demonstrate their applicability.

·       The title of the manuscript is probably not perfectly consistent with its content, since multiple approaches are analysed, and the main innovative contribution is not related to the techniques or algorithms (for image processing or filtering) but rather to proposed configurations and the results of the observability analysis. Probably more focus in the title could be placed on the observability analysis which provides the interesting results for the readers.

Author Response

Please, find the comments and response in the attached document.

Reviewer 2 Report

The manuscript presents a cooperative visual-based SLAM approach for the class of multi-UAV systems that considers a lead agent. In this kind of systems, a team of aerial robots flying in formation must follow a dynamic lead agent, which can be another aerial robot, vehicle or even a human. The manuscript should attract an audience in the scientific field of remote sensing applications. The manuscript is written quite good, but the results must be extended. In Introduction, authors explained the proposed used technology, methods and describe in detail their experiments .

However, I have some important remarks to the manuscript:

Introduction is written well, but should be extended to describe the use of SLAM methods applied to UAVs.

Please describe what kind of objects are supposed to be landmarkrs? In the whole experiment this description has not been included.

I have also a few questions:

Why has not expert work been carried out, even simple research on real low altitude images?

Why does the research experiment only contain numerical calculations without experiments performed on the image?

I have objections to the discussion section. The authors need to re-organize, the results and discussion therein to better highlight to the reader what was done and what is relevant. The gain of the presented technique for the addressed application should be made more explicit in the form: What do the findings allow what was possible before. Authors should discuss the results and how they can be interpreted in perspective of previous studies and of the working hypotheses.

In general, I think paper requires improvement in the context of expanding the experiment.

The article also requires minor editorial corrections.

Author Response

(The authors gave the same response as above.)

Reviewer 3 Report

This paper is well organized and presented what the authors want to study, develop, and argue.

However, the 'simple' simulation result in Chap. 6 is not enough to show an effectiveness and validation of the proposed cooperative visual SLAM algorithm.

Therefore, it is highly recommended to run an experiment with hardware setup in a real environment like some of authors already in the previous work below.

Munguía, R.; Urzua, S.; Bolea, Y.; Grau, A. Vision-Based SLAM System for Unmanned Aerial Vehicles. Sensors 201616, 372.

Author Response

Please, find the comments and response in the attached document.

Round  2

Reviewer 1 Report

I would like to thank the authors for the effort in answering all the comments and suggestions proposed at the first round of review.

The quality and clarity of the manuscript has been significantly improved.

An  additional minor comment is provided below

Lines 161-163. With regards to the answer to this point, the assumption made by the authors is fine. However it would have been nice mentioning this problem (visual detection and tracking of a cooperative UAV) in the introduction, as one of the technologies needed to operate the architectures proposed in this manuscript. Also, not all the algorithms mentioned in the new cited references are based on cooperative, highly-recognizable features installed on board the target UAV. Rather, you are mentioning some non-cooperative approaches for applications like sense-and-avoid which are not consistent with the topic of this manuscript. In this respect, the authors could mention some works specifically tailored to the problem of cooperative UAV tracking. Some possible recent references are suggested below

- Minaeian, S., Liu, J., & Son, Y. J. (2016). Vision-based target detection and localization via a team of cooperative UAV and UGVs. IEEE Transactions on systems, man, and cybernetics: systems, 46(7), 1005-1016.

- Opromolla, R., Fasano, G., & Accardo, D. (2018). A Vision-Based Approach to UAV Detection and Tracking in Cooperative Applications. Sensors, 18(10), 3391.

- Tang, Y., Hu, Y., Cui, J., Liao, F., Lao, M., Lin, F., & Teo, R. S. (2019). Vision-Aided Multi-UAV Autonomous Flocking in GPS-Denied Environment. IEEE Transactions on Industrial Electronics, 66(1), 616-626.

Author Response

Please, find attached the response letter.

Reviewer 2 Report

This manuscript has been improved much and generally I am satisfied with the revisions by the authors.

Author Response

Thanks for your comments and suggestions, the paper has been largely improved thanks to your recommendations.

Reviewer 3 Report

The manuscript is ready for publication.

Author Response

(The authors gave the same response as above.)
